## OPEN

# Neutrophils direct preexisting matrix to initiate repair in damaged tissues

Adrian Fischer[1,8], Juliane Wannemacher[1,8], Simon Christ[1,8], Tim Koopmans[2], Safwen Kadri[1], Jiakuan Zhao[1], Mahesh Gouda[1], Haifeng Ye[1], Martin Mück-Häusl[1], Peter W. Krenn[3,4], Hans-Günther Machens[5], Reinhard Fässler[3], Philipp-Alexander Neumann[6], Stefanie M. Hauck[7] and Yuval Rinkevich[1✉]

Internal organs heal injuries with new connective tissue, but the cellular and molecular events of this process remain obscure. By tagging extracellular matrix around the mesothelium lining in mouse peritoneum, liver and cecum, here we show that pre-existing matrix was transferred across organs into wounds in various injury models. Using proteomics, genetic lineage-tracing and selective injury in juxtaposed organs, we found that the tissue of origin for the transferred matrix likely dictated the scarring or regeneration of the healing tissue. Single-cell RNA sequencing and genetic and chemical screens indicated that the preexisting matrix was transferred by neutrophils dependent on the HSF–integrin AM/B2-kindlin3 cascade. Pharmacologic inhibition of this axis prevented matrix transfer and the formation of peritoneal adhesions. Matrix transfer was thus an early event of wound repair and provides a therapeutic window to dampen scaring across a range of conditions.

njured tissues are replaced by rigid anatomies through accrual of extracellular matrix. These rigid structural and mechanical continuums allow survival of the organism. When normal repair fails, the result is either non-healing chronic wounds or aggravated scarring and fibrosis[1–3]. Impaired healing and excessive scarring are a tremendous burden for people and for the global healthcare system[4,5]. Understanding the repair process and the connective tissue (matrix) accrual is therefore critical to restore and preserve the normal functions of injured adult organs. The primary agents of connective tissue accrual are the fibroblasts, which communicate with other cell types[6–11]. Connective tissue accrual is thought to ensue when a specialized population of fibroblasts migrates into the wound bed to locally synthesize and deposit matrix at sites of injury.

Here we explored the provenance of connective tissue accrual in internal wounds by tagging the preexisting connective tissues underneath the mesothelium surrounding the peritoneum, liver and cecum in mice and following the accumulation of matrix in laparotomy closures, brushing of organ surfaces, ischemic pockets, lipopolysaccharide (LPS) injections and liver electroporations. We found that injury triggered whole-organ mobilization of connective tissue matrix that accrued in wounds and fueled fibroblast activation and scar formation.

## Results

**Preexisting matrix is transferred across organs to seed wounds.** Because loose connective tissue (matrix) serves as the major source for dermal scars in skin[12], we investigated the involvement of preexisting matrix in internal organ injury. For this, we locally tagged and fate-mapped the matrix lining the organ surfaces of live mice using

a nontoxic N-hydroxysuccinimide ester fluorescein (NHS-FITC) (Extended Data Fig. 1a). NHS-FITC labeling did not induce cell death or immune cell recruitment at steady state (Extended Data Fig. 1b,c) and the NHS-FITC+ matrix remained stationary over 24 hours in healthy livers of adult mice (Fig. 1a). To test the mobility of the matrix during injury, we used a wound model based on irreversible electroporation, which is used to operate in various tumor settings. Electroporation kills hepatocytes, and the consequent repair response regenerates without scar tissue[13–15]. We marked the matrix at six distinct locations across the liver, creating circular patches of NHS-FITC+ matrix with clearly defined boundaries that remained stationary in the absence of a secondary injury (Fig. 1a). When a seperate, non-labeled liver location was damaged by electroporation, pools of NHS-FITC+ matrix from all six labeled sites moved from their original location into the injury site within 24 hours postinjury (p.i.), completely filling the wound (Fig. 1a). Three-dimensional imaging of the wounds at 24 hours p.i. showed that the transferred NHS-FITC+ matrix within the wound was structurally distinct from the matrix rigid frames formed by second harmonic generating (SHG) matrix structures (Fig. 1b). At the extremities of the wound, the transferred NHS-FITC+ matrix protruded into filaments that adhered to and wrapped around the SHG+ rigid matrix frames, and interconnected with the adjacent healthy connective tissue (Supplementary Movie 1). Notably, unlabeled, injured surfaces lacked fluorescence (Extended Data Fig. 1d), excluding the possibility of autofluorescence as signal artifacts. These observations indicated that liver injury induced organ-wide transfer of matrix towards the injury site. To test whether transfer of matrix was detected in other organs, we also looked in the

[1]Institute of Regenerative Biology and Medicine, Helmholtz Zentrum München Deutsches Forschungszentrum für Gesundheit und Umwelt (GmbH), München, Germany. [2]Hubrecht Institute,, Developmental Biology and Stem Cell Research, Utrecht, the Netherlands. [3]Department of Molecular Medicine, Max Planck Institute of Biochemistry, Martinsried, Germany. [4]Department of Biosciences and Medical Biology, Paris-Lodron University Salzburg, Salzburg, Austria. [5]Technical University of Munich, School of Medicine, Klinikum rechts der Isar, Department of Plastic and Hand Surgery, Munich, Germany. [6]Technical University of Munich, School of Medicine, Klinikum rechts der Isar, Department of Surgery, Munich, Germany. [7]Metabolomics and Proteomics Core and Research Unit Protein Science, Helmholtz Zentrum München Deutsches Forschungszentrum für Gesundheit und Umwelt (GmbH), München, Germany. [8]These authors contributed equally: Adrian Fischer, Juliane Wannemacher, Simon Christ. ✉e-mail: yuval.rinkevich@helmholtz-muenchen.de

peritoneum. Circular regions of peritoneal matrix were labeled with NHS-FITC, and local injury was induced by brushing a nearby area (Fig. 1c). NHS-FITC[+] matrix transferred into wounds as early as 30 minutes p.i. In a third injury model, peritonea were injured by clinical incision (laparotomy), followed directly by NHS-FITC labeling of six distinct regions of peritoneal matrix (Fig. 1d). At 24 hours p.i., the incision sites were completely occluded with transferred NHS-FITC[+] matrix (Supplementary Movie 2)

Next, we tracked the kinetics of matrix transfer at 24 hours, 72 hours and 2 weeks following liver and peritoneal tissue injury by quantifying the NHS-FITC[+] matrix in the injured sites compared with unwounded tissue that acted as a control. NHS-FITC[+] matrix from the labeled patches expatriated continuously for 2 weeks (Fig. 1e), while the amount of NHS-FITC[+] matrix increased incrementally in both the liver and peritoneum wounds (Fig. 1f), indicating that matrix transfer contributed to formation of new tissue and subsequent remodeling. Multiphoton microscopy showed that NHS-FITC[+] matrix had crosslinked into mature fibrillar connective tissue that patched wounds with scars in peritoneas (Fig. 1g) and livers (Extended Data Fig. 1e) at week 2 p.i. (Supplementary Movies 3–5).

To quantify the ratio of transferred to de-novo-synthesized matrix in wounds, we intraperitoneally (i.p.) injected the non-canonical amino acids (ncAAs) alanine (azidohomoalanine) and glycine (homopropargylglycine), which are randomly integrated into all newly synthesized proteins, including ECM, and can be detected and quantified using simple click chemistry[16], into mice 2 hours before liver and peritoneum injury, directly followed by NHS-FITC labeling of specific liver and peritoneum patches and daily ncAAs injections (Extended Data Fig. 2a). At 72 hours, the ncAAs signal in the muscle tissue acted as control. Robust integration and massive influx of ncAAs[−] NHS-FITC[+] matrix was detected in peritoneal wounds, and staining with an antibody to PDGFR, a marker of fibroblastic cells, did not detect active fibroblasts (Extended Data Fig. 2a), indicating absent or marginal synthesis of new ECM at this time point. Significant amounts of PDGFR[+] fibroblasts could be detected at week 2 p.i. (Extended Data Fig. 2b), and newly deposited ncAAs[+] ECM was woven into and integrated into the transferred NHS-FITC[+] matrix at this time point (Extended Data Fig. 2b).

Large numbers of PDGFR[+] cells are detected in the liver during homeostasis[17]. Although electroporation increased the number of PDGFR[+] fibroblasts in liver wounds at day 3 and week 2 p.i. (Extended Data Fig. 2c–e), injury resulted in the regeneration of the connective tissue, rather than the formation of the fibrotic scars seen in laparotomy closures (Extended Data Fig. 2b). Collagen type 1 staining indicated that the NHS-FITC[+] matrix accounted for 80% of the collagen type 1 protein in healed liver and peritoneal wounds, whereas NHS-FITC[−] collagen accounted for only 20% (Fig. 1g). These experiments indicate that matrix accrual initiated from transferred matrix, and this was gradually remodeled with de-novo-synthesized matrix over time.

**Transferred matrix regenerates connective tissue in wounds.** Next, to investigate how transferred matrix was crosslinked into fibrils in the wound during repair, we labeled live mouse liver surfaces at two distinct locations, one with NHS-EZ-LINK-Biotin and the other with NHS-FITC-ester. After detergent-rich washes to remove fragile interactions, we purified and quantified crosslinked proteins on streptavidin beads. We detected a steady increase in NHS-FITC in the pulldown samples from wounds at 2, 6, 24 and 72 hours p.i. (Fig. 2a), indicating that matrix that originated at distinct organ sites was crosslinked into the repairing wounds to form mature, stably interconnected matrix.

Next, to investigate the distance over which matrix could be transferred, we used a surgical adhesion model in which local abrasion of peritoneal and cecal organ surfaces results in fibrous adhesions between them at 4 weeks p.i. Nearby labeling of the peritoneum (with NHS-FITC) and cecum (with NHS-AF568) indicated cecal and peritoneal matrix intermixing at the injury site, where bands of fibrous adhesions developed 4 weeks p.i. (Fig. 2c). At 2 weeks p.i., pulled-down wound lysates (performed as above) contained abundant NHS-FITC[+] cecal and NHS-EZ-LINK[+] peritoneal proteins (Fig. 2d), indicating that abundant crosslinking had occurred between cecal and peritoneal matrix elements.

To test whether matrix transferred between the peritoneum and liver, we induced adhesion between these two organs by local abrasion of the peritoneum and liver electroporation. We detected transfer and intermixing of NHS-FITC[+] peritoneal and AF568[+] liver matrix at adhesion sites at 4 weeks p.i. (Fig. 2e). Peritoneal-derived NHS-FITC[+] matrix transferred into liver wounds, whereas liver AF568[+] matrix was not detected in the peritoneal wounds (Fig. 2e), indicating that fibrous adhesions and scars originated from the peritoneal but not from the transferred liver matrix (Fig. 2e and Supplementary Movie 6). In injuries induced across two opposite liver lobes, one pre-labeled with NHS-FITC and the other with NHS-PB, we did not detect intermixing of labels or the formation of fibrous adhesions between the lobes at 4 weeks p.i. (Fig. 2f), indicating that only the peritoneal matrix had the ability to form stable scars remotely. Our data suggest that organ-specific matrices formed fibrous adhesions by crosslinking events.

**Transferred matrix predetermines tissue repair outcomes.** Next, we used mass spectrometry to determine the protein constituents of the transferred matrix. We tagged pools of matrix with modified biotin-conjugated EZ-LINK sulfo-NHS esters in the liver, peritoneum and cecum and induced injury at sites distinct from the labeled patches. Twenty-four hours p.i., we collected matrix from the wounds and the labeled sites, purified the EZ-LINK[+] matrix proteins through streptavidin pulldown and performed proteomics of all tagged peptides. We detected hundreds of ECM proteins that represented components of adventitial and serosal connective tissue layers (Extended Data Fig. 3a and Supplementary Tables 2–4) and that could be classified as collagens, ECM glycoproteins, ECM regulators, ECM-affiliated proteins, proteoglycans and secreted

**Fig. 1 | Surface injury induces organ-wide matrix transfer. a**, Representative images from a liver surface marked with NHS-FITC at 24 hours p.i. Scale bar, 2,000 μm. **b**, Representative images of transferred matrix from the original patch and at a distal wound site. Wound area depicted in yellow. Scale bars: wound, 50 μm; original patch, 15 μm. **c**, Representative images of fate mapping of peritoneal surface ECM after brushing injury. Scale bars: overview, 1,000 μm; high magnification, 100 μm. **d**, Representative images showing that peritoneal surface ECM flows towards the laparotomy site. Scale bars: stereomicroscope, 2,000 μm; multiphoton, 15 μm. **e**, Fluorometric measurements of transferred matrix from original patch sites over time. n = 4 biological replicates (C57BL/6J wild-type (WT) mice). Data represent mean ± s.d. One-way ANOVA was used for the multiple comparisons testing, with Tukey's test: *P < 0.05. **f**, Fluorometric measurements of transferred matrix into distal wound sites over time. n = 3 biological replicates (0.5 h, 2 h, 24 h, 2 weeks) and n = 4 (no wound, 6 h, 72 h, 1 week), (C57BL/6J WT mice). Data are mean ± s.d.. Two-tailed Mann–Whitney: *P < 0.05. n.s., P = 0.3429. **g**, Laparotomy closure in matrix fate mapping after 2 weeks. Scale bar: overview, 50 μm; high magnification, 15 μm. Percentage of FITC[+] collagen I at the distal laparotomy site after 2 weeks. Scale bar, 50 μm. Data represent mean ± s.d. Two-tailed Mann–Whitney: *P < 0.05. Data in **a–d** and **g** were selected as representative of six biological replicates (C57BL/6J WT mice) and three independent experiments.

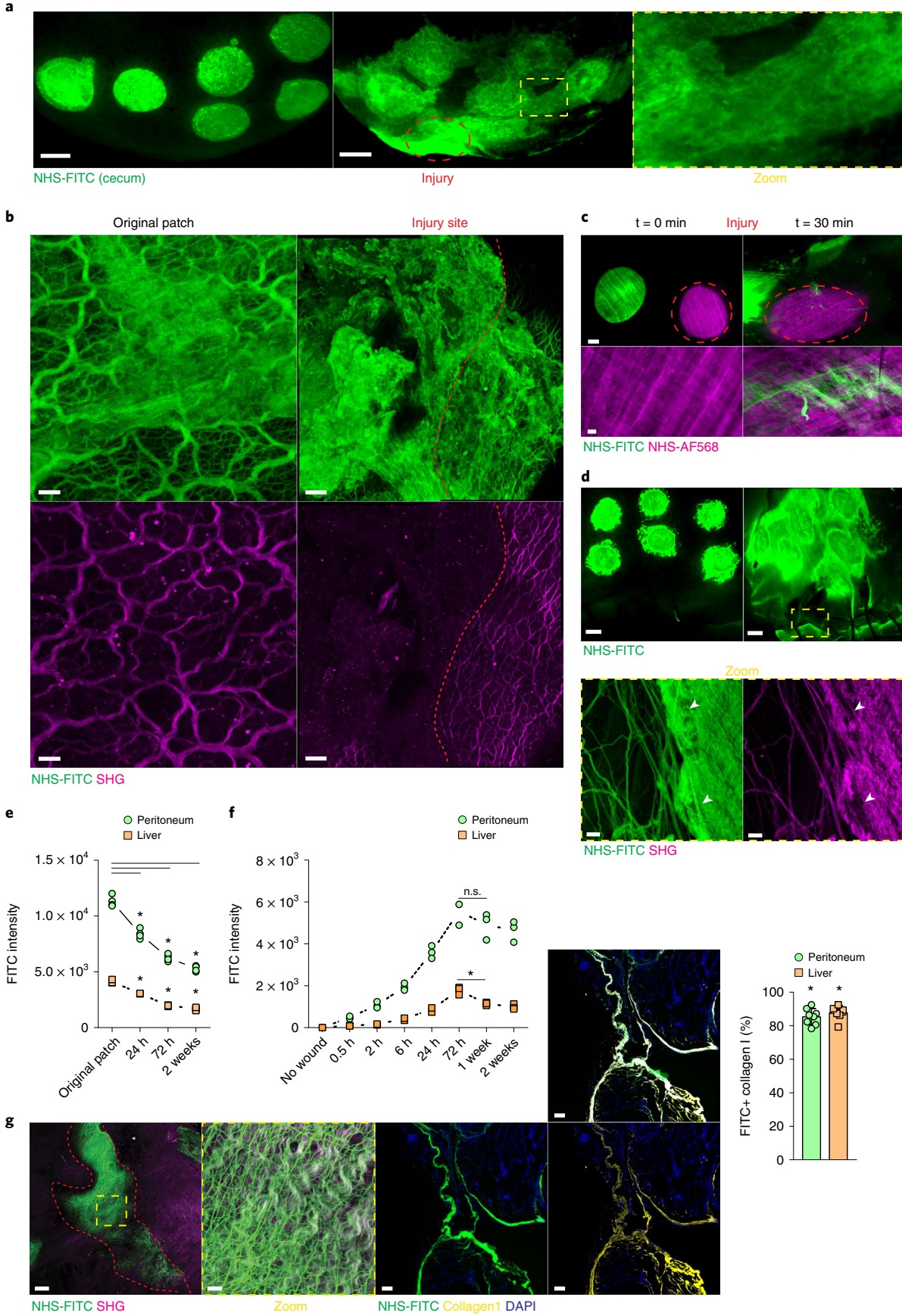

a
NHS-FITC (cecum) Injury Zoom

b
Original patch Injury site
NHS-FITC SHG

c
t = 0 min Injury t = 30 min
NHS-FITC NHS-AF568

d
NHS-FITC
Zoom
NHS-FITC SHG

e
FITC intensity
○ Peritoneum
□ Liver
1.5 × 10⁴
1.0 × 10⁴
5.0 × 10³
Original patch  24 h  72 h  2 weeks

f
FITC intensity
○ Peritoneum
□ Liver
8 × 10³
6 × 10³
4 × 10³
2 × 10³
0
No wound  0.5 h  2 h  6 h  24 h  72 h  1 week  2 weeks
n.s.

○ Peritoneum
□ Liver
FITC+ collagen I (%)
100
80
60
40
20
0

g
NHS-FITC SHG  Zoom  NHS-FITC Collagen1 DAPI

factors[18]. The total matrix pools had a different composition in the liver, peritoneum and cecum, and these organ-specific matrix compositions appeared both in the labeled sites and in the wounds (Fig. 3a). The peritoneum was characterized by higher amounts of collagen, liver had more abundant ECM regulators and the cecal matrix had increased amounts of secreted protein factors (Fig. 3a). We detected abundant fibrillar collagenous fiber proteins, such as collagen type I and III and their crosslinking enzymes, including lysyl oxidase and transglutaminases, which are involved in connective tissue remodeling and maturation (Fig. 3b); proteins involved in basement-membrane formation and stability, such as collagen type IV and VI, including laminins; and elastic fibers (Fig. 3b). This protein inventory contributed to tissue remodeling, fiber clot formation, fibrinolysis, granulation and scar-tissue formation (Fig. 3b).

Principal component analysis of sample distribution indicated that the composition of the transferred matrix was organ-specific, enriched for either scarless repair in liver or scarring and fibrosis in the peritoneum (Fig. 3c). For example, transferred NHS-EZ-LINK+ liver matrix was enriched in regulators of scarless tissue repair, such as Ambp, Itih1, Kng1 and PZP (Fig. 3d), all of which support cellular growth and repair, as well as regulators of oxidative stress, metabolic enzymes and lipid metabolism (Fig. 3d). The NHS-EZ-LINK+ matrix that had originated from the peritoneum was profibrotic and was enriched in collagenous fibers and ECM glycoproteins that induce connective tissue organization, maturation and scar formation (Fig. 3c), including collagen types IX, X and XI, and chaperones of fibrotic scar formation, such as Grem1, Ogn, Chad, MMP9 and MMP20 (Fig. 3). These proteins were completely absent from the liver matrix (Fig. 3e).

To investigate whether these findings translate to humans, we took samples from people of different ages and sexes ($n = 10$) who had developed postoperative adhesions between different abdominal organs, and we used immunofluorescence to determine the protein composition of these adhesions. We detected the same adventitial protein elements found in the mouse peritoneal matrix, such as Lam, Col4, FBN and HSPG2 (Extended Data Fig. 3b), suggesting that human wound repair mobilized matrix from remote adventitial and serosa sites, as in mice. These analyses suggest that the precise protein repertoire in each organ dictated the ensuing repair response, whether scarless or scarring.

**Transferred matrix originates in the mesothelium.** Next, to identify the cellular source of the transferred matrix, we virally transfected an area on the surface of peritoneum or liver with a native collagen 1-binding protein reporter (CNA35) fused to mCherry fluorescent protein (CNA35-mCherry) (Extended Data Fig. 4a). In this reporter system, transduced cells incorporate Col1CNA35-mCherry in the collagen helices and enable real-time visualization and quantification of collagen deposition in live

tissues[19]. Five days after viral transduction, we labeled the same area of the matrix with NHS-FITC, and subjected remote, non-transduced and non-tagged areas to wounding. Light-sheet images (Extended Data Fig. 4b and Supplementary Movie 7) and histology sections (Extended Data Fig. 4c) of the wounded areas indicated extensive accumulation of CNA35-mCherry+NHS-FITC+ matrix in the wounds 24 hours p.i. CNA35-mCherry+NHS-FITC+ matrix made up to 70% and 80% of total collagen in peritoneal and liver wounds, respectively (Extended Data Fig. 4d). Immunohistochemistry staining of the labeled site at week 1 p.i. showed significant amounts of PDPN+ mesothelial cells expressed phosphorylated SMAD2 and SMAD3, indicative of active TGF-β signaling, compared with at 24 hours p.i. (Extended Data Fig. 4e). These experiments indicated that mesothelium was the source of the matrix that was transferred into the wounds.

In skin, a specialized population of En1-lineage positive fibroblasts (EPFs) repairs deep skin wounds by mobilizing distal fascia connective tissue, and mobilization occurs through a collective cell migration that is dependent on N-cadherin in a swarm-like behavior mediated by cell-to-cell contact[12,20]. To explore whether internal organs transferred matrix in a similar way, we labeled and injured peritoneal and liver surfaces as above, and stained wounds at day 1 p.i. with the fibroblast markers PDGFR and NCAD, as well as the neutrophil marker Ly6G. We detected minimal PDGFR+ fibroblasts or N-cadherin expression, and the accumulation of Ly6G+ myeloid cells in wounds (Extended Data Fig. 5a), suggesting fibroblasts were not involved in the transfer of matrix in the liver and peritoneum. To further test this, we used En1CreR26mTmG mice, in which all fascia fibroblasts are GFP+. At 24 hours following liver and peritoneal injuries in the En1CreR26mTmG mice, we did not detect scar-forming GFP+ EPFs in or around the wounds, although we could detect transferred NHS-AF647+ matrix (Extended Data Fig. 5b), suggesting that matrix transfer in liver and the peritoneum was different from the fibroblast-dependent matrix transfer in the skin. Next, we performed liver and peritoneal injuries in C57BL/6J wild-type mice injected with exherin, a blocking peptide that inhibits N-cadherin and blocks fibroblast and fascia movements in skin[20]. N-cadherin blocking had no effect on the transfer of matrix or wound healing in liver or peritoneum at 24 hours p.i. (Extended Data Fig. 5c). These observations suggest that fibroblasts were not involved in the transfer of matrix to wounds in these internal organs.

**Neutrophils carry fibrotic matrix into wounds.** To investigate the link between matrix transfer and inflammation, both of which occur during the early phases of the repair response, we i.p. injected LPS, which is known to induce inflammation-driven fibrosis[21], in mice with local peritoneal and cecal NHS-FITC labeling. Seven days post-LPS injection, we detected massive amounts of NHS-FITC+ matrix within the fibrotic sites in the peritoneum

**Fig. 2 | Transferred matrix regenerates connective tissues in wounds. a**, Fluorometric analysis of samples derived from liver wounds distantly labeled with EZ-LINK-biotin and NHS-FITC at 2, 6, 24 and 72 hours p.i.; uninjured tissue acted as control. Data represent mean ± s.d. Two-tailed Mann–Whitney: *$P < 0.05$. $n = 4$ biological replicates (C57BL/6J WT mice) and 3 independent experiments. **b**, Representative images of a postoperative adhesion site between peritoneum (NHS-AF568+) and cecum (NHS-FITC+) at 4 weeks p.i. Both organs were locally labeled and brushed at distal sites opposing each other. $n = 5$ biological replicates (C57BL/6J WT mice) and 4 independent experiments. **c**, Representative immunolabel images of a postoperative adhesion site between peritoneum (NHS-AF568+) and cecum (NHS-FITC+) at 4 weeks p.i. Both organs were locally labeled and brushed at distal sites opposing each other. $n = 5$ biological replicates (C57BL/6J WT mice) and 4 independent experiments. Scale bars: overview, 100 μm; zoom, 10 μm. **d**, Fluorometric analysis of samples derived from adhesion samples between peritoneum (EZ-LINK-biotin+) and cecum (NHS-FITC+) at 24 hours, 5 days, 2 weeks and 4 weeks p.i.; uninjured peritoneum acted as control. Data represent mean ± s.d. Two-tailed Mann–Whitney; *$P < 0.05$. $n = 5$ biological replicates (C57BL/6J WT mice) and 3 independent experiments. **e**, Representative images of a postoperative adhesion site between peritoneum (NHS-FITC+) and liver (NHS-AF568+) at 4 weeks p.i. Both organs were locally labeled and brushed at distal sites opposing each other. $n = 5$ biological replicates (C57BL/6J WT mice) and 4 independent experiments. Scale bars: overview, 200 μm; zoom, 20 μm. **f**, Representative images of NHS-FITC+ and NHS-PB+ liver lobes with opposing wound sites at 4 weeks p.i. $n = 5$ biological replicates (C57BL/6J WT mice) and 3 independent experiments. Scale bar, 50 μm. Intraorgan adhesions were scored according to Supplementary Table 1. Data represent mean ± s.d. Two-tailed Mann–Whitney: *$P < 0.05$.

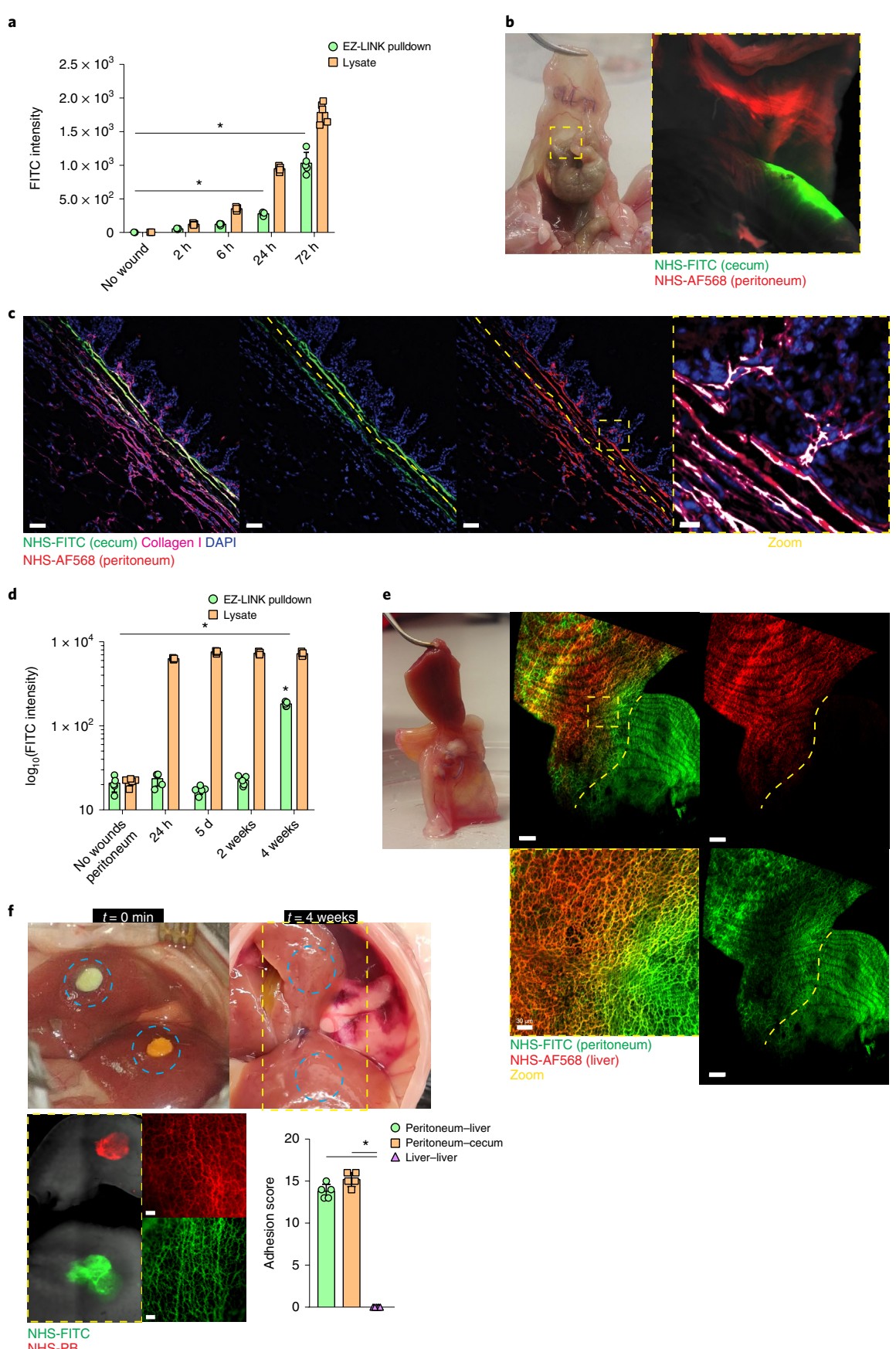

a

b

NHS-FITC (cecum)
NHS-AF568 (peritoneum)

c

NHS-FITC (cecum) Collagen I DAPI
NHS-AF568 (peritoneum)

Zoom

d

e

NHS-FITC (peritoneum)
NHS-AF568 (liver)
Zoom

f

t = 0 min    t = 4 weeks

NHS-FITC
NHS-PB

Peritoneum–liver
Peritoneum–cecum
Liver–liver

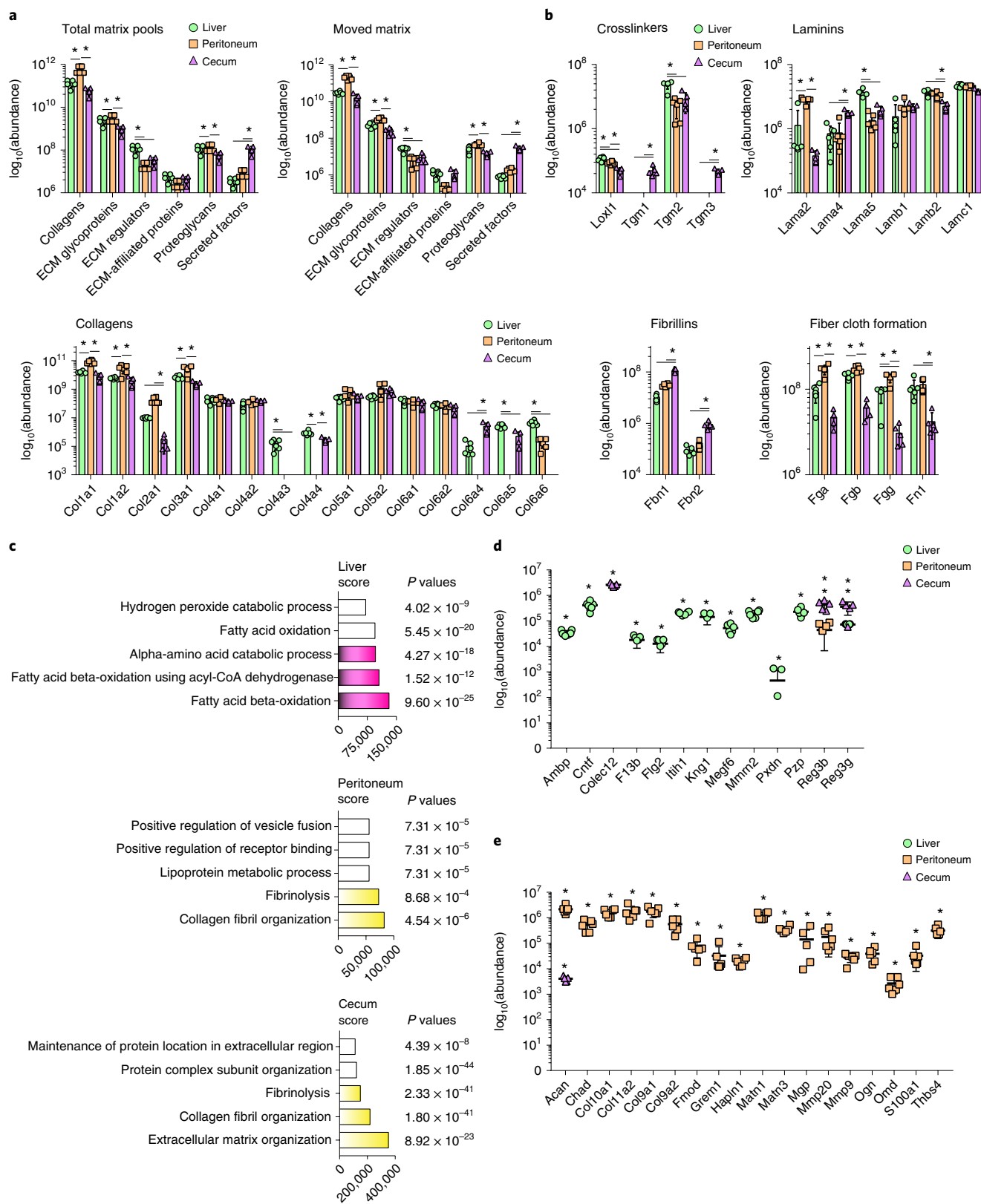

**Fig. 3 | Transferred matrix provides many raw components for tissue repair. a**, Mass-spectrometry analysis of matrisomal proteins derived from pulled-down NHS-EZ-Link marked organ surface ECM. Total matrix pools represent combined total amounts of original label and wound site at 24 hours p.i. Data represent mean ± s.d. One-way ANOVA was used for multiple comparisons testing, with Tukey's test: $*P < 0.05$. **b**, Highlighted matrisomal protein groups derived from pulled down NHS-EZ-Link in wounds 24 hours p.i. from **a**. **c**, Gene ontology analysis of matrisomal genes via EnrichR webtool. Data represent the combined score. $P$ values of Fisher's exact tests are indicated. **d,e**, Matrisomal proteins classified on the basis of Uniprot entries in proregenerative and profibrotic categories. Data represent mean ± s.d. One-way ANOVA was used for multiple comparisons testing, with Tukey's test: $*P < 0.05$. $n = 6$ biological replicates for peritoneum and liver, $n = 5$ biological replicates for cecum (all C57BL/6J WT mice); 3 independent experiments.

and cecum, compared with in mice injected with PBS as control (Extended Data Fig. 6), indicating matrix mobilization occured during sterile inflammation. To investigate the role for myeloid cells in the transfer of matrix, we used Lyz2-Cre Ai14 transgenic mice, in which the tdTomato fluorescent protein is expressed in all cells in the myeloid lineage. Live imaging of liver and peritoneal wounds in Lyz2-Cre Ai14 mice with remote NHS-FITC labeling indicated that tdTomato+ myeloid cells accumulated in wounds at 24 hours p.i. by migrating across large sweeps of organ surfaces (Fig. 4a). We found most, if not all, tdTomato+ myeloid cells (~90%) that migrated from the NHS-FIT-labeled site carried fragments of NHS-FITC+ matrix across organ surfaces (Supplementary Movies 8 and 9). tdTomato+ myeloid cells carrying NHS-FITC+ matrix had no cytoplasmic overlap of the red and green signal (Fig. 4b), suggesting that myeloid cells did not synthesize or phagocytose the matrix. We found NHS-FITC+ matrix was associated with individual tdTomato+ myeloid cells, but also with groups of tdTomato+ myeloid cells that clumped in foci near the wound (Supplementary Movies 10 and 11). These foci contained swarms of active neutrophils, as indicated by staining for CD63, CD66b, FPR1 and TNF receptor (Fig. 4c). Depletion of neutrophils by injection of a Ly6g-neutralizing antibody completely blocked the transfer of NHS-FITC+ matrix in wounds (Fig. 4d), whereas chemical depletion of macrophages with clodronate had no significant effect on matrix transfer (Fig. 4d).

To test whether neutrophils transferred matrix by mediating protease-dependent tissue permeability[22,23], we performed matrix fate mapping in C57BL/6J wild-type mice injected i.p. with inhibitors of aminopeptidase, cathepsin and metalloproteases 1–3, 7–9 and 12–14, all linked to neutrophil functions, or DMSO as control 2 hours before liver and peritoneal injury in conjunction with distant NHS-FITC labeling. Inhibitor treatment had no significant effect on the transfer of NHS-FITC+ matrix or accumulation of Ly6G+ cells in wounds 24 hours p.i. However, inhibition of elastase with either elastinal or elastase inhibitor II significantly increased FITC+ matrix transfer and Ly6G+ cell abundance in wounds compared with in DMSO-treated controls (Fig. 4e), indicating an inhibitory effect of elastases in neutrophil-dependent ECM transfer. In addition, high-resolution images did not detect interstitial or vascular neutrophils (Extended Data Fig. 7a), suggesting matrix transfer was independent of vascular permeability. We did not detect CD45+Ly6G+ circulating neutrophils associated with NHS-FITC+ material in the blood of experimental mice at 24 hours p.i (Extended Data Fig. 7b), indicating that transport of matrix occurred primarily within the interstitial spaces of injured organs. Transfer of NHS-FITC+ matrix into wounds still occurred in the presence of the collagen-synthesis inhibitors tranilast and halofuginone (Fig. 4f), further suggesting the transfer of pre-existent matrix. Neutrophils can modulate fibroblasts to deposit matrix or can degranulate[24,25], and as such contribute to the synthesis of matrix in wounds. However, matrix transfer was independent of neutrophil protein synthesis or degranulation (Fig. 5e,f). Neutrophil swarming and chemoattraction are coordinated through different pathways, triggered by leukotriene, lipoxin A4 and chemokines binding to the chemokine receptors CXCR2 and CXCR4, and nitric oxide[26,27]. Blocking or inhibition of CXCR2 and leukotriene receptor activity completely blocked the transfer of NHS-FITC+ matrix into wounds, whereas inhibition of CXCR4 had no significant effect (Fig. 4g). Local application of lipoxin A4 on liver surfaces induced the recruitment Ly6G+ neutrophils and the transfer of NHS-FITC+ matrix at 24 hours p.i., compared with PBS application in the absence of injury (Fig. 4h). Overall, these data indicate that neutrophils transferred preexisting matrix into the sites of injury.

**Neutrophils use integrin Mβ2 to carry matrix into wounds.** To further study the specific cells populating the wounds, we used highly parallel single-cell transcriptomics of 25,000 cells isolated at day 1 and day 7 p.i. from the wounds and uninjured control livers. This analysis differentiated 17 distinct immune cell populations within the wound (Fig. 5a,b), of which myeloid cells, such as monocytes and macrophages, represented the predominant lineage (55%) (Fig. 5a–c). Next, we determined the expression of membrane receptors that might enable the transport of extracellular matrix to wound sites in the myeloid, including neutrophil clusters (Fig. 5d). We noticed gene expression in neutrophils was altered during the liver injury response, with a significant upregulation of the integrin subunit αM (ITGAM) compared with in neutrophils from healthy tissue (Fig. 5e). Integrins ITGAM and ITGB2 heterodimerize to form αMβ2 (also known as Mac-1, CD11b/CD18 or CR3).

Analysis of inferred lineage relationships among neutrophils indicated that neutrophils displayed substantial heterogeneity between day 1 and day 7 in pairwise correlation of cells as well as in hierarchical clustering of genes. Particularly, at 24 hours p.i., increased expression of genes like CXCR4 or CD24a (Fig. 5f,g) indicated the gradual maturation and differentiation of neutrophils over time, while increased expression of apoptotic factors[27] and age-related genes[28] (Fig. 5h) indicated their activation in response to electroporation. Pseudotemporal positioning of neutrophils revealed a temporal sequence of gene regulatory events during neutrophil maturation, ending at 7 days p.i. with multiple end points indicative of diverse subpopulations (Fig. 5i,j), suggesting that neutrophils differentiated into defined subpopulations after injury within the wound, ultimately resulting in apoptosis.

Single-cell sequencing data showed there was no significant upregulation of matrix-associated genes in neutrophils during liver injury (Fig. 5k,l), indicating that neutrophils do not synthesize matrix. Immunolabeling in C57BL/6J wild-type mice detected the upregulation of ITGAM and ITGB2 on neutrophils in peritoneal and liver wounds at 24 hours p.i. (Fig. 6a). I.p. injections with neutralizing antibodies against ITGAM and ITGB2 2 hours before matrix labeling in C57BL/6J mice led to a reduction or complete cessation of NHS-FITC+ matrix transfer in the wound at 24 hours p.i. compared with a control antibody (Fig. 6b), indicating that matrix

---

**Fig. 4 | Active neutrophils actively transport matrix by swarming. a**, Snapshots of Supplementary Movies 7 and 8: Lyz2+ (red) cells on liver are shown, and peritoneal surfaces single cells are highlighted with arrows. $n = 5$ biological replicates and 3 independent experiments. Two-tailed Mann–Whitney: *$P < 0.05$. Scale bars, 50 μm. **b**, Representative images of NHS-FITC-labeled liver and peritoneal surfaces in Lyz2Cre;Ai14 mice at 24 hours p.i. $n = 5$ biological replicates and 3 independent experiments. Scale bars: overview, 50 μm; zoom, 5 μm. **c**, Representative images showing the ratio of TNF-α+, FPR1+, CD63+, CD62L+ and CD66b+ and Ly6G+ neutrophils in wound areas at 24 hours p.i. $n = 6$ biological replicates (all C57BL/6J WT mice) and 4 independent experiments. Scale bar: 50 μm. **d**, Representative images from liver surfaces of animals treated with anti-Ly6G antibody or clodronate marked with NHS-FITC at 24 hours p.i. $n = 5$ biological replicates (all C57BL/6J WT mice) and 3 independent experiments. Two-tailed Mann–Whitney: *$P < 0.05$. Scale bars: overview, 500 μm; histology, 30 μm. **e–g**, Fluorometric measurements of transferred FITC+ matrix in wounds in animals treated with protease (**e**), collagen-synthesis (**f**) or neutrophil-swarming inhibitors (**g**) at 24 hours p.i. Quantification of Ly6G+ cells in wounds at 24 hours p.i. $n = 5$ biological replicates (all C57BL/6J WT mice) and 3 independent experiments. Two-tailed Mann–Whitney: *$P < 0.05$. **h**, Representative images from liver surfaces of animals locally treated with lipoxin 4A marked with NHS-FITC at 24 hours p.i. Quantification of Ly6G+ cells in wounds at 24 hours p.i. $n = 4$ biological replicates (all C57BL/6J WT mice) and 4 independent experiments. Two-tailed Mann–Whitney: *$P < 0.05$. Scale bar: overview, 500 μm.

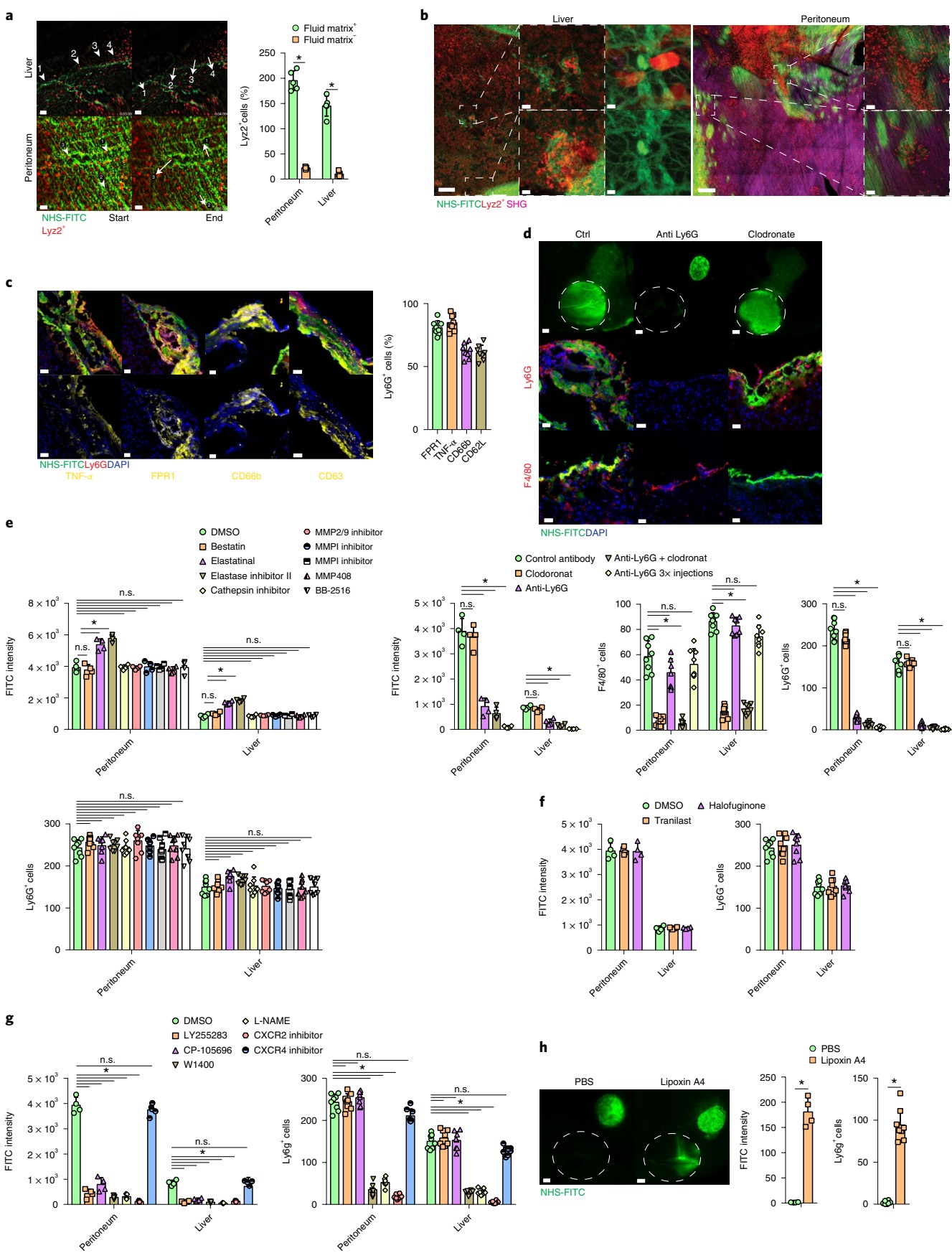

transfer was dependent on integrin activation. Although transferred FITC+ matrix could not be detected after αMβ2 blocking, Ly6G+ neutrophils were still present in the wounds of these mices compared with those that received control antibody injections (Fig. 6b), suggesting that integrin blocking inhibited matrix transport, and not neutrophil migration, extravasation or crawling into wounds.

Kindlin3 is an essential component of integrin-mediated focal adhesions that form between cells and the extracellular matrix[28,29]. To explore how integrin-dependent signaling in neutrophils regulated matrix transfer, we used R26[CreER]Kindlin3[fl/fl] mice, in which the integrin signaling-mediator kindlin3 can be conditionally deleted by administration of tamoxifen, resulting in a block in integrin signal transduction. To address whether matrix transfer into wounds depended on intact integrin signaling (that is, intact kindlin3 expression), we adoptively transferred neutrophils from tamoxifen-treated R26[CreER]Kindlin3[fl/fl] mice (kindlin3− neutrophils) and control R26[CreER-negative]Kindlin3[fl/fl] mice (kindlin3+ neutrophils) into wild-type mice in which neutrophils had been depleted by intravenous injection of Ly6G antibodies, followed by liver and peritoneal injury after neutrophil reconstitution (Fig. 6c). NHS-FITC+ matrix was not detected in the wounds of Ly6G antibody-treated mice in the absence of neutrophil reconstitution (Fig. 6c), but accumulated in mice reconstituted with kindlin3+ neutrophils (Fig. 6c). Transferred Ly6G+kindlin3− neutrophils were detected in wounds in large numbers at 24 hours p.i., but NHS-FITC+ matrix was not detected (Fig. 6c). Thus, integrin signaling in neutrophils was required for matrix transfer into wounds.

**Neutrophilic response to injury is HSF dependent.** To establish the upstream signals of matrix transfer into wounds, we investigated the role of 'heat shock' signaling, which is a known pharmacological target in fibrosis[30,31] and has been shown to regulate integrin function[32,33]. I.p. injection of the heat shock factor (HSF) inhibitors HSI and KRIBB11 2 hours before injury blocked NHS-FITC+ matrix accumulation 24 hours p.i. in liver and peritoneal wounds to the same extent as neutrophil depletion or integrin signaling inhibition in kindlin3− neutrophils (Fig. 7a). Next, we asked whether heat shock signaling controlled integrin activation in neutrophils. αMβ2 heterodimerization is a critical stage in integrin activation[20]. More ITGB2 immunoprecipitated with ITGAM in lysates of wound-derived neutrophils 24 hours p.i. compared with blood neutrophils from non-wounded mice (Fig. 7b). Moreover, injection of HSI and KRIBB11 decreased αMβ2 immunoprecipitated in wound-derived neutrophils (Fig. 7b), indicating heat shock proteins induce heterodimerization.

To investigate the clinical role of HSF in wound response, we took peritoneal biopsies from R26[CreER]Kindlin3[fl/fl] mice, labeled the ECM ex vivo with NHS-FITC and cultured these tissues for 24 hours with neutrophils pretreated with HSF inhibitor 1 or DMSO as control (Extended Data Fig. 8a). NHS-FITC+ matrix transfer was not observed in tissues cultured with HSF inhibitor1-treated neutrophils, in contrast with DMSO-treated neutrophils (Extended Data Fig. 8a). Next we intraperitoneally transferred neutrophils pretreated with HSF inhibitor 1 or DMSO into tamoxifen-treated R26[CreER]Kindlin3[fl/fl] (Kindlin3 cKO) mice directly after matrix tagging and peritoneal or liver injury. Both DMSO- and HSF-inhibitor-1-treated Ly6G+ neutrophils were detected in wounds 24 hours p.i. (Fig. 7c). However, NHS-FITC+ matrix was only transferred in Kindlin3 cKO mice reconstituted with DMSO-treated neutrophils (Fig. 7c).

We investigated whether heat shock protein inhibition could be used as a pharmacological intervention against matrix transfer and peritoneal adhesions (scars). We induced surgical adhesions between peritoneum and cecum and applied a single dose of 10 nM HSI or DMSO i.p. We detected transfer of NHS-FITC+ matrix into sites of adhesion as early as 24 hours p.i. in mice treated with DMSO, but not with in mice treated with HSI (Fig. 7d). Four weeks p.i., peritoneal adhesions in mice treated with DMSO were filled with a foundation of NHS-FITC+ collagen I protein (Fig. 7d), indicating that it originated in remote locations. In the presence of transferred NHS-FITC+ matrix, peritoneal adhesions incorporated active PDGFR+YAP/TAZ+ fibroblasts and pSMAD2/3+PDPN+ mesothelial cells (Fig. 7d), indicative of remodeling and maturation and mature long-lasting adhesions. Conversely, application of HSI completely blocked the transfer of NHS-FITC+ matrix (Fig. 7d), while scars and long-lasting adhesions failed to form by 4 weeks p.i., without detectable impairment of wound closure or healing (Fig. 7d). These results indicate the therapeutic potential of blocking HSF-integrant signaling in order to prevent matrix transfer, scars and peritoneal adhesions in mice.

## Discussion

Here we show that organ connective tissues contained mobile matrix reservoirs, and that injury triggered organ-wide transfer of this preexisting matrix into injured tissue, where they fueled tissue repair. Premade connective tissue matrix moved from the mesothelial layer surrounding visceral and parietal internal organs, across the organ, into injured sites. This immature proteinaceous matrix was crosslinked to reestablish rigid connective tissue frames to repair breaches in liver or peritoneum. We found that neutrophils had an essential role in transferring matrix into wounds through heat shock–integrin signaling.

Although our results indicated that mesothelium exported bulk connective tissue into wounds, this is not mutually exclusive to the prevailing view that fibroblasts deposit matrix to further consolidate scars over extended periods of time in response to a much stiffened and biomechanically altered wounded microenvironment[34]. We have demonstrated here that this altered biomechanical, physical and signaling scar environment was likely provided by imported matrix. Indeed, we have shown that in the absence of matrix transfer, wounds failed to incorporate active fibroblasts or to mature into long-lasting scars. Fibroblasts are therefore part of a secondary

**Fig. 5 | Single-cell transcriptomics of liver wounds identifies 17 major cellular lineages. a,** Single-cell sequencing data of adult mouse livers subjected to irreversible electroporation, showing distribution of cells per cluster, colored by experimental condition. **b,** UMAP of **a**. **c,** Scaled heatmap (yellow, high; purple, low) of cell-lineage markers used to identify the different cell populations. **d,** Scaled heatmap (yellow, high; purple, low) of all ECM receptors expressed in the different liver lineages. **e,** Violin plots showing the expression of Itgb2 and Itgam in the neutrophil cluster. Two sample t-test. **f,** Inferred lineage relationship among neutrophils in an adjacency network on the basis of pairwise correlations between cells. Black arrow indicates point of heterogeneity. **g,** Hierarchical clustering on neutrophilic genes identified by principal component analysis (PCA). Black arrows indicate points of heterogeneity. **h,** Aging and apoptosis score based on the total expression of age-related genes (listed in **d**) and apoptosis genes (GO:0097193). **i,** Scaled heatmap (yellow, high; purple, low) of all age-related genes (see Methods). **j,** Monocle pseudotemporal ordering based on genes identified by PCA revealing neutrophilic maturation. **k,** Cumulative expression score of genes that fall under the core matrisome, further categorized into collagens (top), glycoproteins (middle) and proteoglycans (bottom). **l,** Mesenchyme cell cluster, showing gene expressions (violin plots) of all collagens exceeding an expression threshold of 2.0. ***$P < 0.001$. Box plots represent the median, interquartile range (IQR), minimum (25th percentile, 1.5 × IQR) and maximum (75th percentile, 1.5 × IQR).

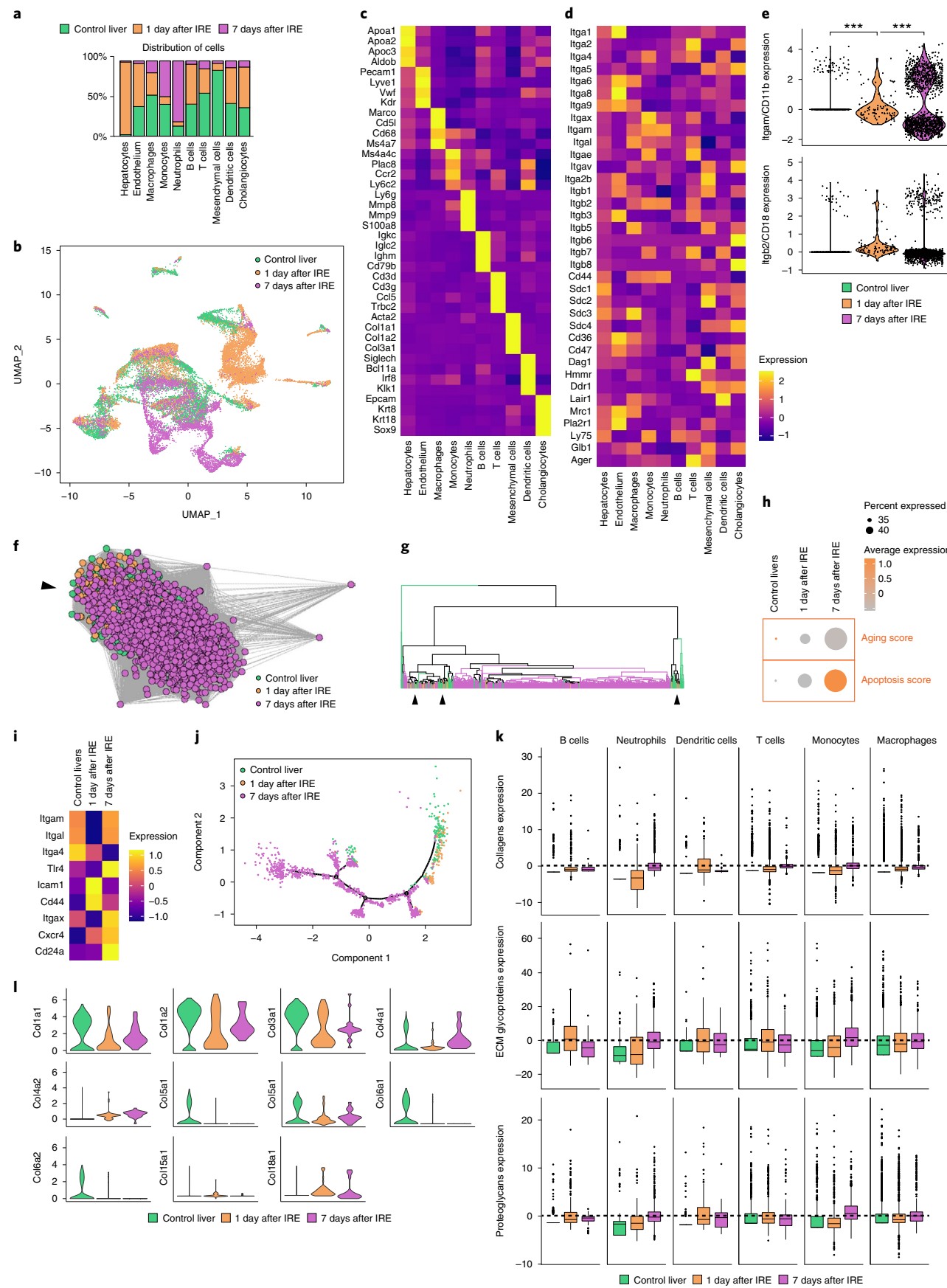

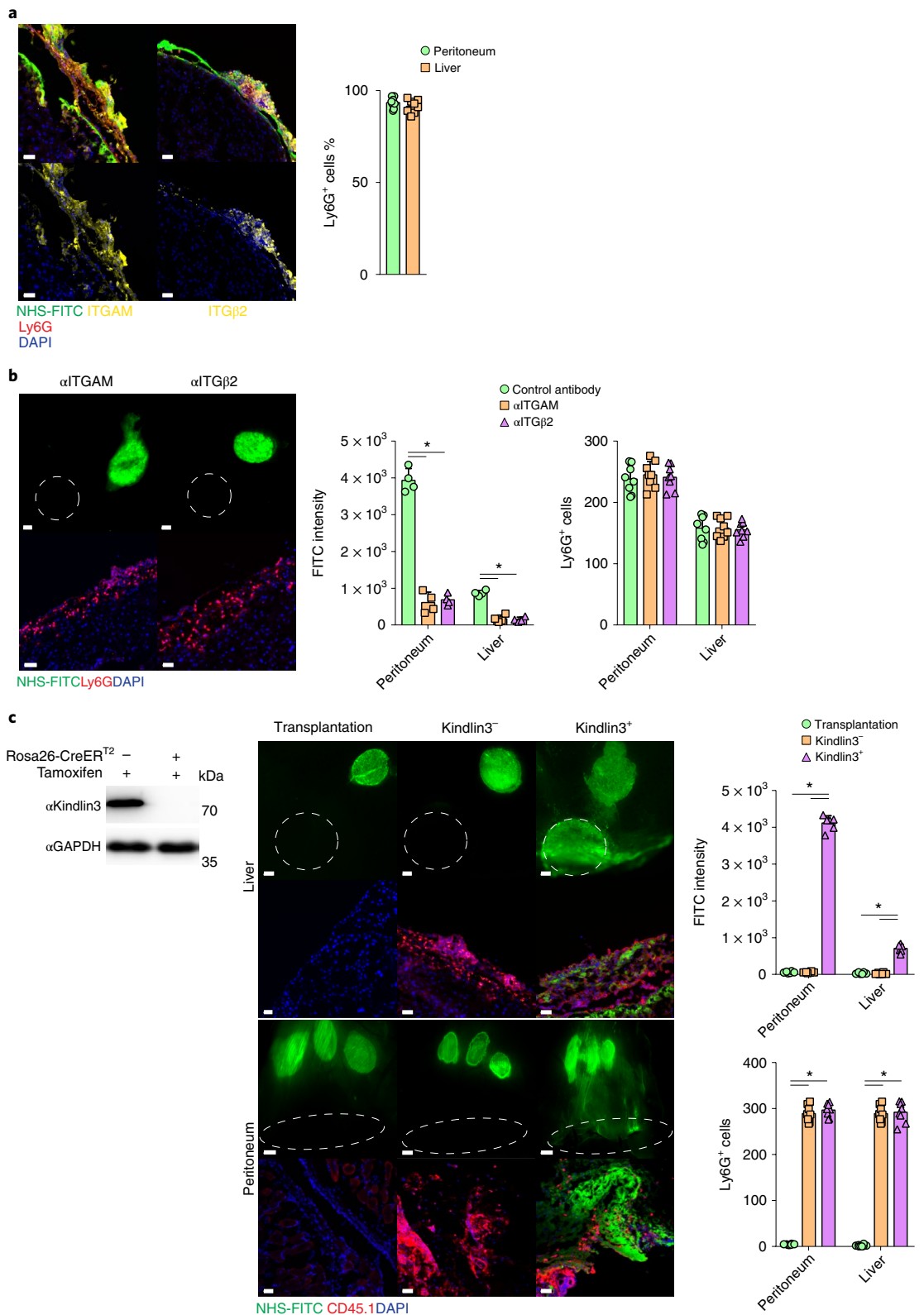

**Fig. 6 | Integrin AM and B2 in neutrophils orchestrate matrix transfer. a**, Representative immunohistological stainings of organ wounds 24 hours after injury. $n = 6$ biological replicates (C57BL/6J WT mice). Data are representative of 4 independent experiments. Histology, 50 μm. Two-tailed Mann–Whitney: *$P < 0.05$. **b**, Representative images from liver surfaces of animals treated with ITGAM- and ITGβ2-blocking antibodies marked with NHS-FITC at 24 hours p.i. and quantification of Ly6G+ cells in wounds post 24 hours. $n = 6$ biological replicates (C57BL/6J WT mice). Data are representative of 4 independent experiments. Two-tailed Mann–Whitney: *$P < 0.05$. Scale bar: overview, 500 μm; histology, 30 μm. **c**, Representative images from organ surfaces of neutrophil-depleted animals marked with NHS-FITC at 24 hours p.i. Purified neutrophils were derived from Cre− or Cre+ R26CreER;floxKindlin3flox mice treated with tamoxifen before transplantation into the abdomen. Successful recombination was verified via immunoblot analysis. $n = 5$ biological replicates of 4 independent experiments. Scale bar: overview, 500 μm; histology, 30 μm.

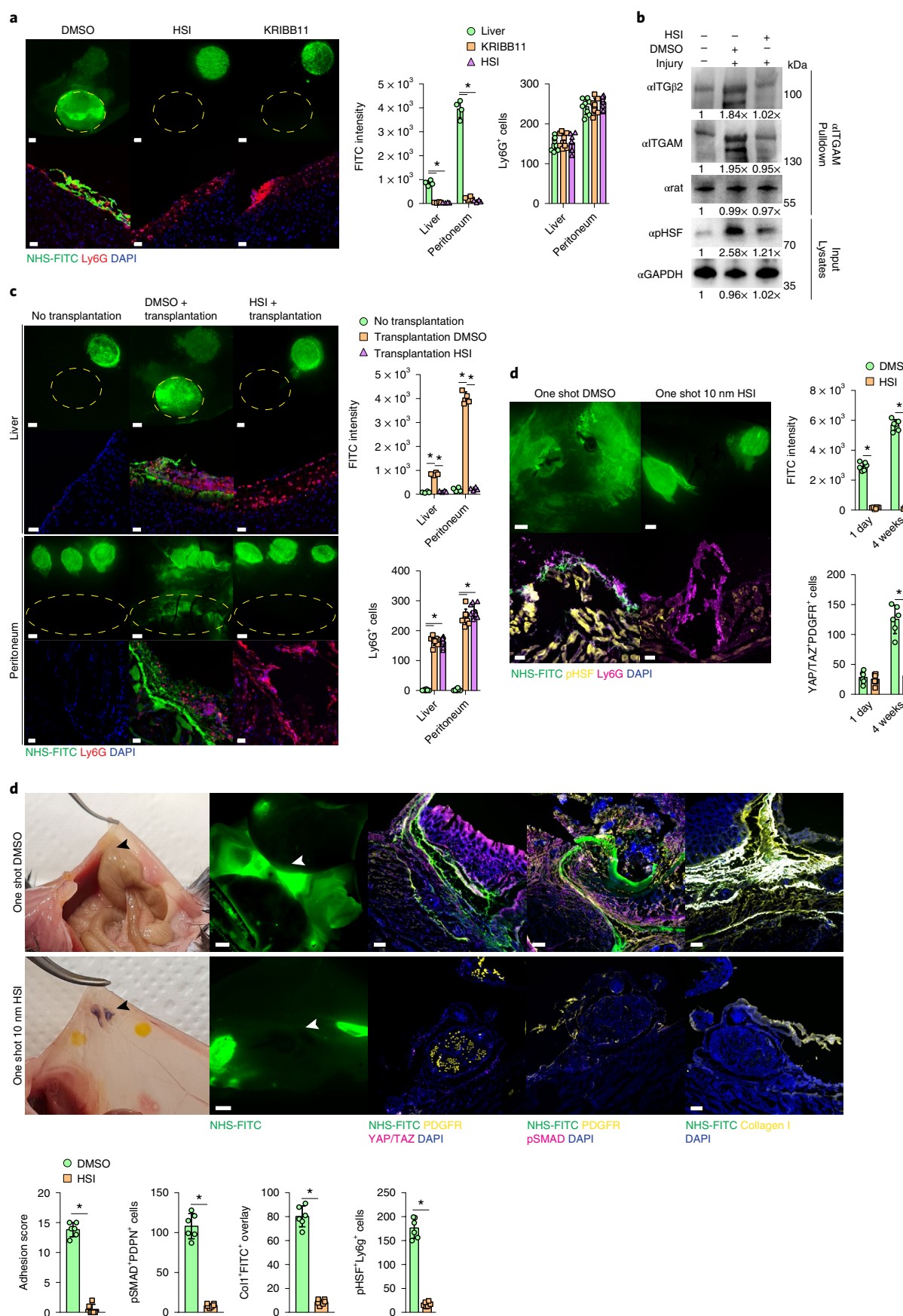

**Fig. 7 | Neutrophils direct matrix transfer. a**, Representative images from liver surfaces of animals treated with HSIs marked with NHS-FITC at 24 hours p.i. *n* = 4 biological replicates (C57BL/6J WT mice) and 3 independent experiments. Two-tailed Mann–Whitney: *\*P < 0.05*. Scale bar: overview, 500 µm; histology, 30 µm. **b**, Representative immunoblots of lysates and immunoprecipitations derived from purified wound neutrophils. *n* = 4 biological replicates (C57BL/6J WT mice) and 3 independent experiments. Quantifications are relative to controls. **c**, Representative images from organ surfaces of neutrophil-depleted animals marked with NHS-FITC 24 hours p.i. Purified neutrophils were pretreated with DMSO or HSI before transplantation into the abdomen. *n* = 4 biological replicates (C57BL/6J WT mice) and 3 independent experiments. Two-tailed Mann–Whitney: *\*P < 0.05*. Scale bar: overview, 500 µm; histology, 30 µm. Scale bar: overview, 500 µm; histology, 30 µm. **d**, Representative images from postoperative adhesions of HSI-pretreated animals with NHS-FITC labeled peritoneum 1 day p.i. *n* = 6 biological replicates (C57BL/6J WT mice) and 4 independent experiments. Two-tailed Mann–Whitney: *\*P < 0.05*. Scale bars: overview, 2,000 µm; histology, 100 µm. **e**, Representative images from post operative adhesions of HSI-pretreated animals with NHS-FITC-labeled peritoneum at 4 weeks p.i. *n* = 6 biological replicates (C57BL/6J WT mice). Two-tailed Mann–Whitney: *\*P < 0.05*; n.s., not significant. Adhesions were scored according to Supplementary Table 1. Data are representative of four independent experiments. Scale bars: overview, 2,000 µm; histology, 100 µm.

response to the initial scar-formation process launched by matrix transfer.

We found that matrix transportation was not exclusive to organ surfaces but could happen in deep interstitial tissue once a deeper incision was made. We speculate that matrix transportation may play a much broader role in wound repair and fibrosis beyond organ surfaces. Importantly, we found around 80% of scar tissue came from distal mobil matrix reservoirs.

Our proteomic data from the mesothelial matrix material transferred into wounds leads us to speculate that the specific protein composition of the transferred matrix, rather than the organ fibroblasts, determine the diverging fibrotic responses that develop during adult tissue repair. Indeed, the repertoire of transferred matrix elements varied between peritoneum, cecum and liver. In the mesothelial lining of the liver, the transferred matrix contained many enzymes and pro-regeneration proteins and injuries that transferred matrix across adjacent liver lobes after injury, leading to scarless repair without fibrous adhesions between the opposing lobes. The idea that preexisting matrix contributes to wound repair was demonstrated by juxtaposing liver and non-regenerating peritoneal tissue and injuring the interface locally. When injured liver was in direct proximity to injured peritoneum surfaces, peritoneal matrix was transferred on the liver, initiating fibrous adhesions and scars between liver and peritoneum.

Understanding the molecular choreography that moves matrix into wounds provides multiple opportunities for therapeutic intervention. Using time-lapse video, we showed that, at 24 hours p.i., tissue repair was directed by neutrophils, which steered mesothelial matrix into tissue-repair sites within minutes. Neutrophils upregulated the collagen-binding integrins αMβ2, which heterodimerized to transfer matrix downstream of HSF signaling.

A link between integrins, kindlin3 and various developmental and wound-healing processes also exists in humans. Leukocyte adhesion deficiency type III (LADIII) is a disorder characterized by dysfunctional kindlin3 activity. People with LADIII syndrome have abnormal bone structure, uncontrollable bleeding and impaired wound healing[35], which is associated with connective tissue deposition. As matrix transport was dependent on kindlin3, we speculate that the LADIII-associated phenotypes may be linked with impaired matrix transport.

Neutrophils have an important and early role in early adhesion formation, primarily through neutrophil extracellular trap (NET) secretion[36]. Here we showed that neutrophils provided the components necessary for fibrotic scar tissue. Transcriptionally distinct neutrophil subpopulations can be distinguished in terms of their maturity and distinct functions[37]. Our analysis did not indicate which distinct population of neutrophils contributed to matrix transport. However, the scRNA-seq data suggested that the matrix movement was accomplished by neutrophils that activated HSF, upregulated ITGAM and ITGβ2, and homed towards wounds through CXCR2, NOS and LTB4R chemoattraction. Once in the wounds, neutrophils underwent further maturation, ending in apoptosis. The exact half-life of neutrophils is still unclear, but recent methods estimate that it is between 6 and 12 hours[38]. Because matrix transport continued for several days and reached a plateau at about 72 hours, we assume that new neutrophils are constantly migrating into the wound during the process. While our scRNA-seq data indicated that different subpopulations of neutrophils entered the wound, future studies are needed to refine which subsets of neutrophils enable matrix transport, or whether matrix transport is universal to all mature subsets. Expression of ITGβ2 on neutrophils seemed to be necessary for matrix transport, but not for the physical relocation of neutrophils into wounds, as ITGβ2 blocking affected matrix cargo, but did not block neutrophil recruitments into wounds. Once transported into wounds, matrix served as a provisional wound material, where it was further crosslinked covalently to other matrix elements. We also found that, as matrix was moved into wounds from remote organ locations, mesothelial cells initiated de novo synthesis of ECM, and replenished matrix pools within the organ.

Heat shock signaling regulates wound healing, presumably through reducing TGF-β signaling and the differentiation of fibroblasts[39–41]. Our data indicate that the HSF–integrin axis acted in neutrophils to transfer matrix into wounds. Moreover, heat shock factors acted as potential pharmacologic targets by blocking matrix seeding in wounds. Modulation of matrix transfer by heat shock factors in neutrophils during early wound repair creates a new therapeutic space to treat impaired wounds and excessive scarring.

## Online content

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

## Methods

**Animals.** All mouse lines (C57BL/6J, B6.129P2-Lyz2tm1(cre)Ifo/J (Lyz2Cre), B6;129S6-$Gt(ROSA)26Sor^{tm14(CAG-tdTomato)Hze}$/J (Ai14)), En1Cre (En1tm2(cre)Wrst) R26mTmG (Gt(ROSA)26Sortm4(ACTB-tdTomato,-EGFP)Luo) were obtained from Jackson Laboratories or Charles River. R26CreER;floxKindlin3flox mice were provided by P. K. and R. F. Animals were bred and maintained in the Helmholtz Animal Facility in accordance with EU directive 2010/63. Animals were housed in individual ventilated cages, and animal housing rooms were maintained at a constant temperature of 20–24 °C and humidity of 45–65% with a 12-hours light cycle and were supplied with water and chow ad libitum. All animal experiments were reviewed and approved by the Government of Upper Bavaria and registered under the project number ROB-55.2-2532.Vet_02-19-133 or ROB-2532.Vet_02-19-148 and conducted under strict governmental and international guidelines. This study is compliant with all relevant ethical regulations regarding animal research.

**Injury models.** Thirty minutes before surgery, mice received a preemptive subcutaneous injection of metamizole (200 mg/kg body weight). Anesthesia was supplied by an intraperitoneal injection of a medetomidine (500 µg/kg), midazolam (5 mg/kg) and fentanyl (50 µg/kg) cocktail, hereafter referred to as MMF. Anesthetic depth was assessed by toe reflex. Eyes were covered with bepanthen-cream to avoid dehydration, and the abdomen was shaved and disinfected with betadine and sterile PBS. Animals were kept on their backs on a heating plate at 39 °C. A midline laparotomy (1–1.5 cm) was performed through the skin and peritoneum. Four hooks, positioned around the incision and fixed to a retractor and magnetic base plate, allowed for clear access to the abdominal cavity and liver.

Local damage to the liver surface was induced via electroporation tweezers by applying 30-V, 50-ms pulses at 1-second intervals for 8 cycles. Before closure of the incision, buprenorphine (0.1 mg/kg) was pipetted in the abdomen to allow for initial postsurgical analgesia. For long-term analgesia, metamizole (novalgin, 200 mg/kg) was provided through daily injections. The peritoneum and skin were closed with two separate 4-0 silk sutures (Ethicon). Upon closure of the incision, mice were woken up by antagonizing medetomidine and midazolam through a subcutaneous cocktail injection of atipamezol (1 mg/kg) and flumazenil (0.25 mg/kg). Mice were allowed to recover on a heating pad, after which they were single-housed. Mice were euthanized after the indicated time points, and liver tissue was obtained. In the peritoneal model, the surgical procedure was as described above, but the peritoneal areas were marked.

To induce adhesions between liver and peritoneum, abrasion and talcum was applied to the electroporated side of the liver and to the opposite side of the peritoneum. In the peritoneal–cecal adhesion model, surfaces of the cecum and peritoneum were injured with a brush, two surgical knots were placed, and talcum powder was applied onto wound sides of both organs. Liver injury was done by electroporation of opposing sites and application of talcum powder. The sterile inflammation model was enacted by locally labeling organ surfaces as described above, and 5 days later repeated injections of sterile lipopolysaccharide in PBS (100 µg/20 g body weight) were performed. To deplete neutrophils, the rat anti-mouse Ly6G monoclonal antibody, clone 1A8 (BioxCell), was injected i.p. 2 days after the i.p. injection of clodronate liposomes (Liposoma). Macrophage depletion was performed with a single i.p. injection of clodronate liposomes. Neutralizing antibodies (Bio X Cell) were applied at a concentration of 200 µg/20 g body weight. CP-105,696 and LY255283 (Sigma Aldrich) were injected i.p. 2 hours before surgery at 10 µM in sterile PBS. Lipoxin (Merck Millipore) was applied locally by soaking the reagent in a sterile filter paper with 100 nM solution and applying the filter paper over the liver surface for 5 minutes. AHA (0.05 mg/g) or HPG (0.025 mg/g) were applied 2 hours prior to surgery and afterwards daily through i.p. injection. Skin wounds were performed according to Correa-Gallegos et al.[12].

**Human tissue.** All human samples were obtained from surgeries at the Department of Surgery, Klinikum rechts der Isar, Technical University of Munich, Germany, following approval of the local ethics committee of the Technical University of Munich, Germany (no. 173/18 S). Adhesions were intraoperatively diagnosed and dissected from the respective organs and prepared for further analysis.

**Labeling of ECM on organ surfaces.** Succinimidyl esters (NHS-esters; Thermo Fisher) were diluted in DMSO to a concentration of 25 mg/ml and stored at −80 °C. To obtain ectopic labeling of matrix, we generated a labeling solution by mixing NHS-ester 1:1 with 100 mM pH 9.0 sodium bicarbonate buffer. Sterile Whatman filter paper (Sigma Aldrich) biopsy punches were soaked in NHS-labeling solution and locally placed on the liver surface. After 1 minute, the labeling punch was removed. For kinetic measurements organ surfaces were marked with a 2-mm filter patch of NHS-FITCproximally (1.0 cm away).

**AAV production.** Production and purification of AAV preparations for AAVCNA35mCherry was performed according to the AAVpro Purification Kit Maxi (Takara Bio) protocol. In brief, 5× T225-flasks were triple-transfected with the plasmid pHelper from the kit AAVpro Helper Free System (AAV6) (Takara Bio), the plasmid pAAV2/8RGD containing coding sequences of the AAV2-derived rep proteins and the modified AAV8 capsid proteins and the pAAV-Cp-SV40pA derivative containing the AAV genome with the respective transgene. At 96 hours post-transfection, cells were collected and AAV vector particles were released by breaking up the cells with 3× freeze–thaw cycles. Genomic DNA was digested with Cryonase cold-active nuclease, and AAV vector particles were separated from cell debris by filtration (0.45-µm filter). Finally, AAV particles were separated from low molecular contaminants by using 100-kDa size-exclusion columns and were concentrated. Titers of final AAV preparations were determined through qPCR utilizing the AAVpro Titration Kit (qPCR) V2 (Takara Bio). Local AAV labeling was performed using sterile whatman filter papers, as described above.

**Tissue preparation.** Upon organ excision, organs were fixed overnight at 4 °C in 2% formaldehyde. The next day, fixed tissues were washed three times in Dulbecco's phosphate buffered saline (DPBS, GIBCO, no. 14190-094), and depending on the purpose, were either embedded, frozen in optimal cutting temperature (OCT) compound (Sakura, no. 4583) and stored at −20 °C, or stored at 4 °C in PBS containing 0.2% gelatin (Sigma Aldrich, no. G1393), 0.5% Triton X-100 (Sigma Aldrich, X100) and 0.01% Thimerosal (Sigma Aldrich, no. T8784) (PBS-GT). Fixed tissues were embedded in OCT compound and cut with a Microm HM 525 (Thermo Scientific) by the standard protocol. In short, sections were fixed in ice-cold acetone for 5 min at −20 °C and then washed with PBS. Sections were then blocked for non-specific binding with 10% serum in PBS for 60 minutes at room temperature, and then incubated with primary antibody in blocking solution overnight at 4 °C. The next day, following washing, sections were incubated in PBS with fluorescent secondary antibody, for 120 minutes at room temperature. Finally, sections were washed and incubated with Hoechst 33342 nucleic acid stain (Invitrogen, no. H1399), washed in ddH₂O, mounted with Fluoromount-G (Southern Biotech, no. 0100-01), and stored at 4 °C in the dark.

Primary antibodies were used at 1:100, were from Abcam, and were raised in rabbit unless otherwise stated. Antibodies were raised against the following epitopes: collagen I (1:150, Rockland), collagen 4, fibronectin, HSPG2 (Elabscience), cleaved caspase 3, laminin, hamster anti-PDPN, rat anti-LY6G, TNF, ITGAM, ITGβ2, CD45.2, FPR1, CD62L, YAP, goat anti-PDGFR, TAZ, pSMAD, goat anti-PDGFR, phosphoHSF (Elabscience) and N-cadherin.

Alexa-Fluor-488-, Alexa-Fluor-568- or Alexa-Fluor-647-conjugated antibodies (1:500, Life Technologies) against suitable species were used as secondary antibodies. H&E staining was performed according to the manufacturer's protocol (Sigma).

Visualization of ncAAs was performed using Alkyne-Alexa Fluor 647 (Thermo Fisher) and Click-iT Cell Reaction Buffer Kit (Thermo Fisher).

**Microscopy.** Histological sections were imaged under a M205 FCA Stereomicroscope (Leica) and ZEISS AxioImager Z2m (Carl Zeiss). For whole-mount 3D imaging of tissues, fixed samples were embedded in 35-mm glass-bottom dishes (Ibidi) with low-melting-point agarose (Biozym) and left to solidify for 30 min. Imaging was performed with a Leica SP8 multiphoton microscope . For time-lapse imaging of liver and peritoneal tissues, samples were embedded as just above. Imaging medium (DMEM/F-12) was then added. Time-lapse imaging was performed under a M205 FCA Stereomicroscope. A modified incubation system, with heating and gas control (ibidi), was used to guarantee physiologic and stable conditions during imaging. Temperature control was set to 35 °C with 5% CO₂-supplemented air. Whole-mount samples for 3D lightsheet imaging were stained and cleared by a modified 3DISCO protocol[42]. Samples were dehydrated in an ascending tetrahydrofuran (Sigma Aldrich) series (50%, 70%, 3×100%; 60 minutes each), and subsequently cleared in dichloromethane (Sigma Aldrich) for 30 min and then immersed in benzyl ether (Sigma Aldrich). Microscopy samples were kept in cooled, light-protected vials in the dark. Cleared samples were imaged while submerged in benzyl-ether with a lightsheet fluorescence microscope (LaVision BioTec). While submerged in benzyl-ether, specimens were illuminated on two sides by a planar light-sheet using a white-light laser (SuperK Extreme EXW-9; NKT Photonics). Optical sections were recorded by moving the specimen chamber vertically in 5-mm steps through the laser light-sheet. Two-, three- and four-dimensional data were processed with Imaris 9.1.0 (Bitplane) and ImageJ (1.52i). Contrast and brightness were adjusted for better visibility. Overlay channels were generated using Imaris 9.1.0. Quantifications were performed with ImageJ.

**Protein biochemistry.** Tissues were snap frozen and ground using a tissue lyser (Qiagen). Pulverized tissues were resuspended in lysis buffer (20 mM Tris-HCl pH 7.5, 1% Triton X-100, 2% SDS, 100 mM NaCl, 1 mM sodium orthovanadate, 9.5 mM sodium fluoride, 10 mM sodium pyruvate, 10 mM beta-glycerophosphate), and supplemented with protease inhibitors (complete protease inhibitor cocktail, Pierce) and kept for 10 minutes on ice. Samples were then sonicated and spun down for 5 minutes at 10,000g. Supernatants were stored at −80 °C. Protein concentrations were determined via BCA assay, according to the manufacturer's protocol (Pierce).

Protein pulldown was as follows. Lysates were diluted with a pulldown buffer (20 mM Tris-HCl pH 7.5, 1% Triton X-100, 100 mM NaCl, supplemented with

protease and phosphatase inhibitors) and incubated overnight with dynabeads (Thermo Fisher) according to the manufacturer's instructions at 4 °C on a rotator. The next day, the samples were each diluted twice with wash buffer 1 (pulldown buffer plus 2% SDS) and then with wash buffer 2 (pulldown buffer with reduced, 0.5% Triton X-100) and were finally washed twice with wash buffer 3 (20 mM Tris-HCl pH 7.5 and 100 mM NaCl). Beads were then resuspended in Elution Buffer (20 mM Tris-HCl pH 7.5, 100 mM NaCl and 50 mM DTT) and incubated for 30 minutes at 37 °C. Finally, the samples were boiled for 5 minutes at 98 °C and the supernatants were stored at −80 °C. Lysates for fluorescence measurements were handled in cooled, light protected vials. Fluorescence intensities of lysates were measured in a Fluostar optima fluorometer (BMGlabtechImmunoprecipitation was as previously described[43]. SDS–PAGE and western blots were performed with the BioRad Wet tank system. Immunoblots were performed via rabbit-anti-GAPDH (1:1,000, Abcam), rabbit-anti-phosphoHSF (1:1,000, Elabscience), rabbit-anti-ITGAM (1:1,000, Abcam) and rabbit-anti-ITGβ2 (1:1,000, Abcam). Anti-rabbit- and anti-rat-HRP was purchased from Biorad and was used at 1:20,000. Quantification of immunoblots was performed using ImageJ.

**Mass spectrometry.** Tissues were marked locally with an EZ-LINK-NHS 100:1 FITC-NHS mixture. After 24 hours, the organs were removed. Tissue pieces from the original marking were separated from moved matrix fractions and snap frozen. Tissue lysis was performed as described above. Samples were digested by a modified FASP procedure[44]. After reduction and alkylation using DTT and IAA, the proteins were centrifuged on Microcon centrifugal filters (Sartorius Vivacon 500 30 kDa), washed thrice with 8 M urea in 0.1 M Tris/HCl pH 8.5 and twice with 50 mM ammonium bicarbonate. The proteins on the filter were digested for 2 hours at room temperature using 0.5 µg Lys-C (Wako Chemicals) and for 16 hours at 37 °C with 1 µg trypsin (Promega). Peptides were collected by centrifugation (10 minutes at 14,000g), acidified with 0.5% TFA and stored at −20 °C until measurements. The digested peptides were loaded automatically onto an HPLC system (Thermo Fisher Scientific) equipped with a nano trap column (100 µm ID × 2 cm, Acclaim PepMAP 100 C18, 5 µm, 100 Å/size, LC Packings, Thermo Fisher Scientific) in 95% buffer A (2% ACN, 0.1% formic acid (FA) in HPLC-grade water) and 5% buffer B (98% ACN, 0.1% FA in HPLC-grade water) at 30 µl/min. After 5 min, the peptides were eluted and separated on the analytical column (nanoEase MZ HSS T3 Column, 100 Å, 1.8 µm, 75 µm × 250 mm, Waters) at 250 nl/min flow rate in a 105-minute nonlinear acetonitrile gradient from 3% to 40% in 0.1% formic acid. The eluting peptides were analyzed online in a Q Exactive HF mass spectrometer (Thermo Fisher Scientific, Bremen, Germany) coupled to the HPLC system with a nano spray ion source and operated in the data-dependent mode. MS spectra were recorded at a resolution of 60,000, and after each MS1 cycle, the 10 most abundant peptide ions were selected for fragmentation. Raw spectra were imported into Progenesis QIsoftware (version 4.1, Nonlinear Dynamics, Waters). After feature alignment and normalization, spectra were exported as Mascot Generic files and searched against the SwissProt mouse database (16,872 sequences) with Mascot (Matrix Science, version 2.6.2) with the following search parameters: 10 ppm peptide mass tolerance and 0.02 Da fragment mass tolerance, two missed cleavages allowed, carbamidomethylation was set as fixed modification, and camthiopropanoyl, methionine and proline oxidation were allowed as variable modifications. A Mascot-integrated decoy database search calculated an average false discovery of <1% when searches were performed with a mascot percolator score cut-off of 13 and an appropriate significance threshold $P$.

Peptide assignments were re-imported into the Progenesis QI software and the abundances of all unique peptides allocated to each protein were quantified/counted. The resulting normalized abundances of the individual proteins were used for calculation of protein ratios and $P$ values (ANOVA) between sample groups using a nested design. Gene ontology analysis was performed using the EnrichR webtool[45,46]. Extracellular elements were identified through a database search against a matrisomal database[18].

**Single-cell RNA-seq.** Three livers per experimental group were pooled for each sequencing run. For each liver, the electroporated area was punched out with a circular 4-mm biopsy punch, and subsequently minced with fine scissors into small pieces (approximately 1 mm²). The equivalent but non-injured area was used in control livers. The resulting fragments were further processed by enzymatic digestion in 5 mL enzyme mix consisting of dispase (50 caseinolytic units/ml), collagenase (2 mg/ml), and DNase (30 µg/ml), for 30 minutes at 37 °C under constant agitation (180 rpm). Enzyme activity was inhibited by adding 5 ml of PBS supplemented with 10% FBS. Dissociated cells in suspension were passed through a 70-µm strainer and centrifuged at 500g for 5 minutes at 4 °C. Red blood cell lysis (Thermo Fisher; 00-4333-57) was performed for 2 minutes and stopped with 10% FBS in PBS. After another centrifugation step, the cells were counted in a Neubauer chamber and critically assessed for single-cell separation and viability. A total of 250,000 cells were aliquoted in 2.5 ml of PBS supplemented with 0.04% of BSA and loaded for Drop-seq at a final concentration of 100 cells/µL. Drop-seq experiments were performed as described previously[47]. In brief, using a microfluidic PDMS device (Nanoshift), single cells were co-encapsulated in droplets with barcoded beads (Chemgenes Corporation) at a final concentration of 120 beads/µL. Droplets were collected for 15 min/sample. After droplet breakage, beads were collected,

washed, and prepared for on-bead mRNA reverse transcription (Maxima RT, Thermo Fisher). Following an exonuclease I (New England Biolabs) treatment for the removal of unused primers, beads were counted, aliquoted (2,000 beads/reaction, equals ~100 cells/reaction), and pre-amplified by 13 PCR cycles (primers, chemistry, and cycle conditions identical to those described previously[47]). PCR products were pooled and purified twice on 0.6× clean-up beads (CleanNA). Prior to tagmentation, cDNA samples were loaded on a DNA High Sensitivity Chip on the 2100 Bioanalyzer (Agilent) to ensure transcript integrity, purity and quantity. For each sample, 1 ng of pre-amplified cDNA from an estimated 1,000 cells was tagmented by Nextera XT (Illumina) with a custom P5 primer (Integrated DNA Technologies). Single-cell libraries were sequenced in a 100-bp paired-end run on an Illumina HiSeq4000 using 0.2 nM denatured sample and 5% PhiX spike-in. For priming of read 1, 0.5 µm Read1CustSeqB was used (primer sequence: GCCTGTCCGCGGAAGCAGTGGTATCAACGCAGAGTAC).

**Blood preparation and data analysis for flow cytometry.** Murine blood was collected by cardiac puncture and diluted in PBS with heparin. Red blood cells were lysed twice in 1× BD Pharm lysing solution. Antibodies against CD45 and Ly6G were mixed with whole cells together with LIVE/DEAD fixable near-IR dead cell staining dye. Neutrophils were identified as CD45⁺Ly6G⁺, excluding doublets and dead cells. Acquisition was performed on BD LSR II (BD) using FACSDiva software (BD), then data were analyzed with FlowJo software (BD, v10.8).

**Statistics.** Data were analyzed in Prism 7.0.0 (GraphPad). Statistical tests were performed as indicated in the figure legends, and $n$ values are also provided. All error bars represent mean ± s.d. Mice and tissues were randomly assigned to treatment groups where applicable. No data were excluded. Data were presumed to be normally distributed. Statistical significance was defined as $P < 0.05$, for reasons of space and visibility, the individual $P$ values have not been integrated into the figures.

**Reporting Summary.** Further information on research design is available in the Nature Research Reporting Summary linked to this article.

## Data availability

Source Data for the figures are provided in the individual source file. Raw files of mass-spectrometry experiments are available at PRIDE accession code: PXD031766. RNA-sequencing datasets generated in this paper are available under GEO accession code: GSE198828. Source data are provided with this paper.

## Code availability

Code used for Massspec processing is available at: https://github.com/kadrisaf/Fluid_Matrix

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

## Acknowledgements

We thank S. Dietzel and the Core Facility Bioimaging at the Biomedical Centre of the Ludwig-Maximilians Universität München for access and support with the multiphoton system. We thank H. Schiller and M. Stamnitz for access and support for scRNA-seq equipment. Y.R. was supported by the Human Frontier Science Program Career Development Award (CDA00017/2016), the German Research Foundation (RI 2787/1-1 AOBJ: 628819), the Fritz–Thyssen–Stiftung (2016-01277), the Else-Kröner-Fresenius-Stiftung (2016_A21) and the European Research Council Consolidator Grant (ERC-CoG 819933). A. F. was supported by PFP – Helmholtz Postdoctoral Fellowship.

## Author contributions

Y.R. supervised the research. A.F., J.W. and Y.R. designed the experiments. A.F. performed animal surgeries with the help of J.W. A.F. designed and performed protein biochemistry experiments. A.F. performed immunofluorescence staining imaging and analysis. H.Y. provided skin tissue samples. A.F. performed stereomicroscope and time-lapse imaging. J.W. provided veterinary advice and prepared animal experiment protocols. S.C. performed multiphoton imaging. T.K. performed single-cell experiments.

A.F., C.M. and H.S. visualized mass-spectrometry data. P.-A.N. provided human adhesion samples. S.M.H. and S.K. analyzed mass-spectrometry experiments. M.G. performed light sheet imaging. J.Z. and P.W.K. purified neutrophils. J.Z. performed FACS analysis. P.W.K. and R.F. provided Kindlin3 conditional knockout mice. M.M.-H. designed and produced CNK35 reporter vectors. HGM provided support and assisted in clinical interpretation of the animal data. A.F., T.K. and Y.R. wrote the manuscript.

## Funding

## Competing interests
The authors declare no competing interests.

## Additional information
**Extended data** is available for this paper at https://doi.org/10.1038/s41590-022-01166-6.

**Correspondence and requests for materials** should be addressed to Yuval Rinkevich.

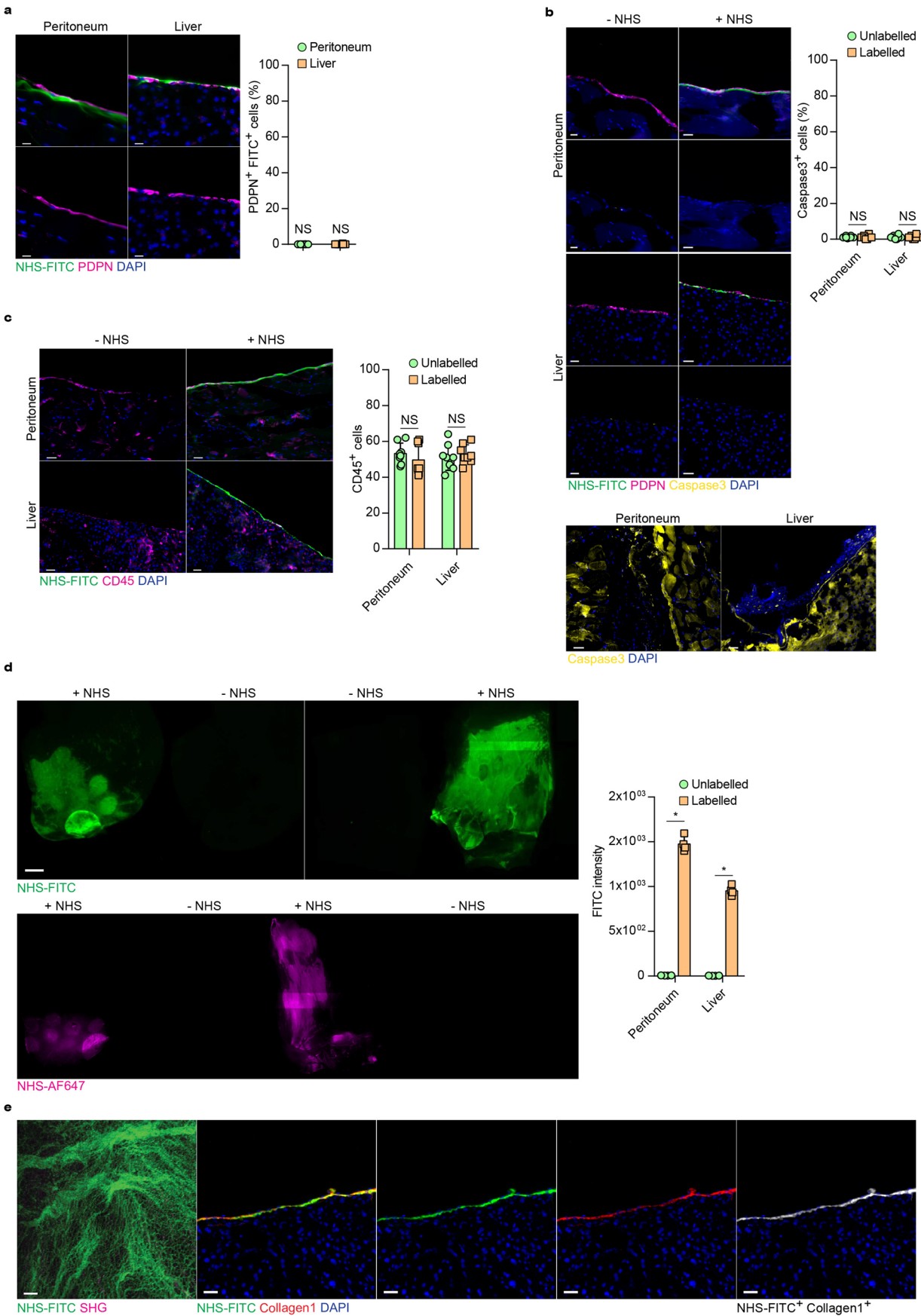

**Extended Data Fig. 1 | See next page for caption.**

**Extended Data Fig. 1 | NHS esters enable stable, non-toxic, labeling of organ surface ECM. a-c**, Representative immunohistological stainings for NHS labelling, cell death (Caspase3) and immune cell (CD45) amount in control and NHS-labeled organ surfaces. Peritoneal closure and liver electroporation acted as positive control for Caspase3 stainings. n = six biological replicates. (C57BL/6 J WT mice) and three independent experiments. **d**, Representative images from liver and peritoneal surfaces of animals locallylabelled with NHS-FITC or NHS-AF647 24 hours post injury. n = four biological replicates. Scale bar: 2000 μm. Fluorometer based analysis of FITC signals detected in wounds 24 hours after injury quantified signal to autofluorescence ratio. n = four biological replicates and three independent experiments. Data represented are mean ± SD. Two-tailed Mann–Whitney; * P < 0.05. **e**, Representative multiphoton and immunohistological images of a liver wound 2 weeks after injury. n = six biological replicates. Scale bar: Multiphoton: 50 μm; Histology: 50 μm. Two-tailed Mann–Whitney; * P < 0.05; ns= not significant. n = six biological replicates. (C57BL/6 J WT mice) and four independent experiments.

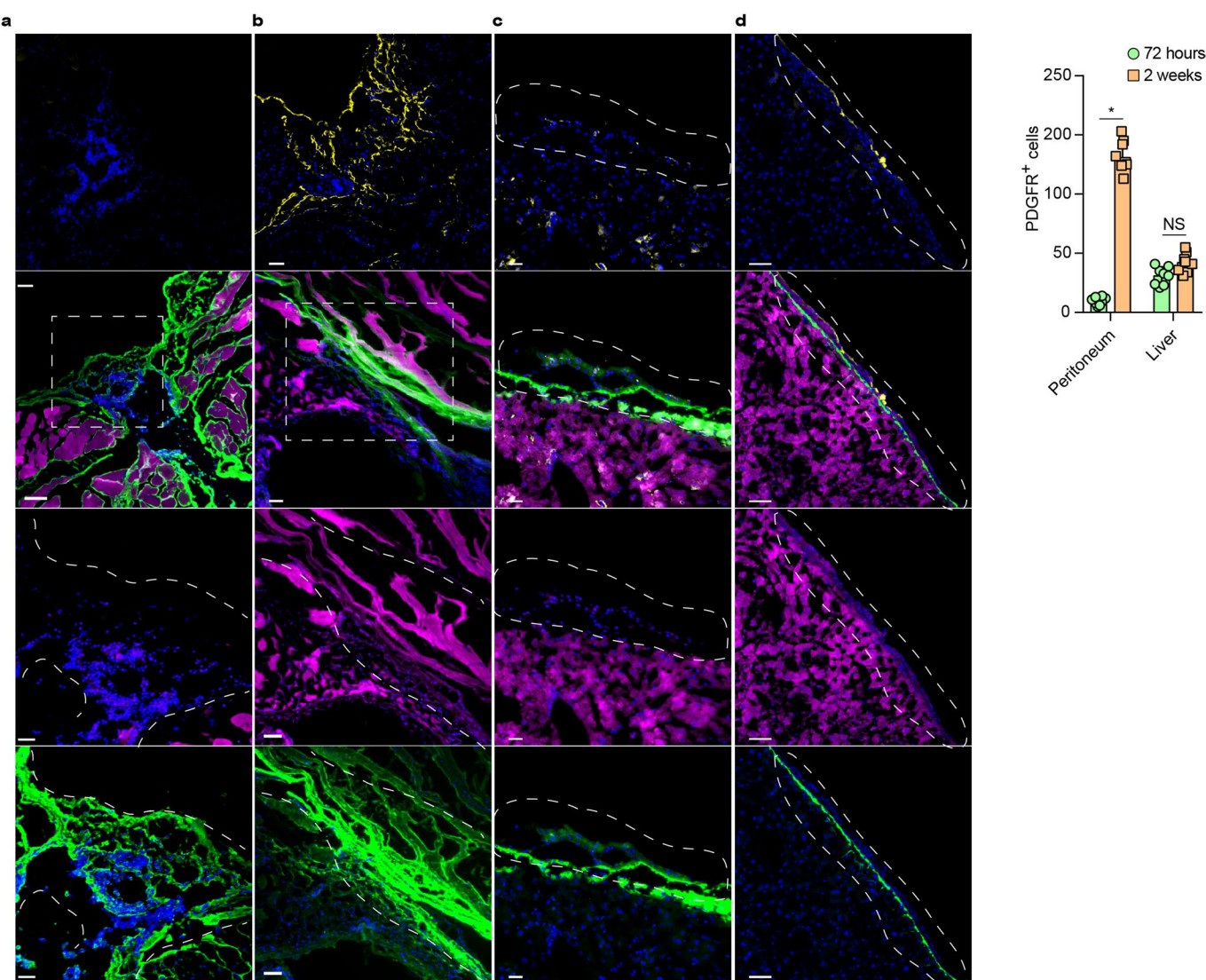

NHS-FITC PDGFR DAPI De novo

**Extended Data Fig. 2 | Fluid matrix mediated wound closure is supplemented days later by de novo ECM synthesis. a-d**, Representative immunohistological images of peritoneal and liver wounds of animals injected with non-canonical amino acids (ncAAS; magenta) and locally labeled with NHS-FITC, samples were collected three days and 2 weeks after injury. n = six biological replicates (C57BL/6 J WT mice) and three independent experiments. Overview: 100 μM; Highlight: 50 μM. Two-tailed Mann–Whitney; * P < 0.05; NS = not significant.

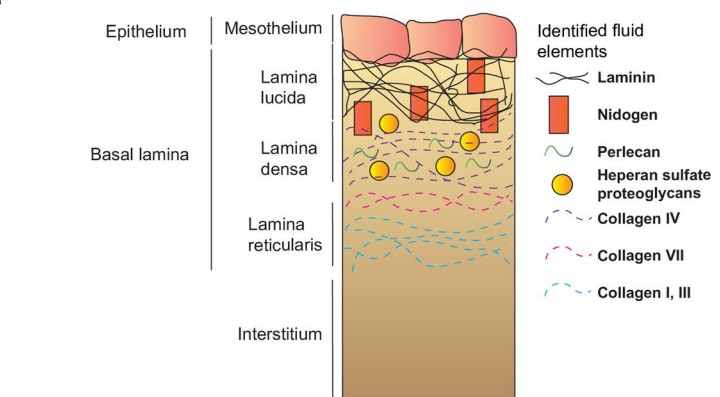

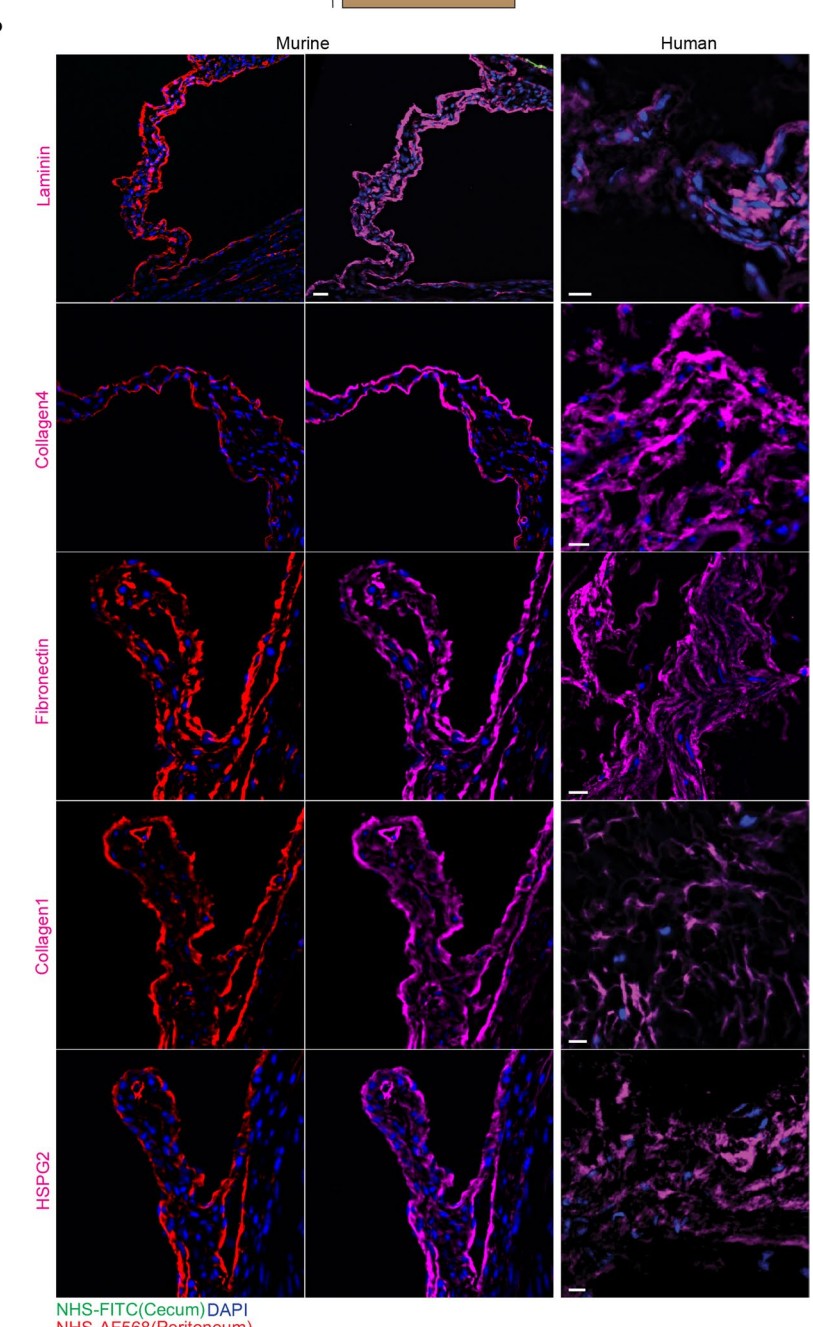

**Extended Data Fig. 3 | See next page for caption.**

**Extended Data Fig. 3 | Fibrous postsurgical adhesions consist of transferred matrix elements. a**, Schematic Visualization of adventitial and serosal connective tissue layers identified transferred matrix. **b**, Representative immunofluorescence images of histological sections of three biological replicates of murine and human abdominal postsurgical adhesions. Murine peritonea were labeled with NHS-AF568 cecum was labeled with NHS-FITC. Mice were sacrificed 4 weeks after surgery. n = six C57BL/6 J WT mice and four independent experiments. Human adhesion tissue was collected from ten different patients with abdominal adhesions during adhesiolysis. Scale bar: 20 μm.

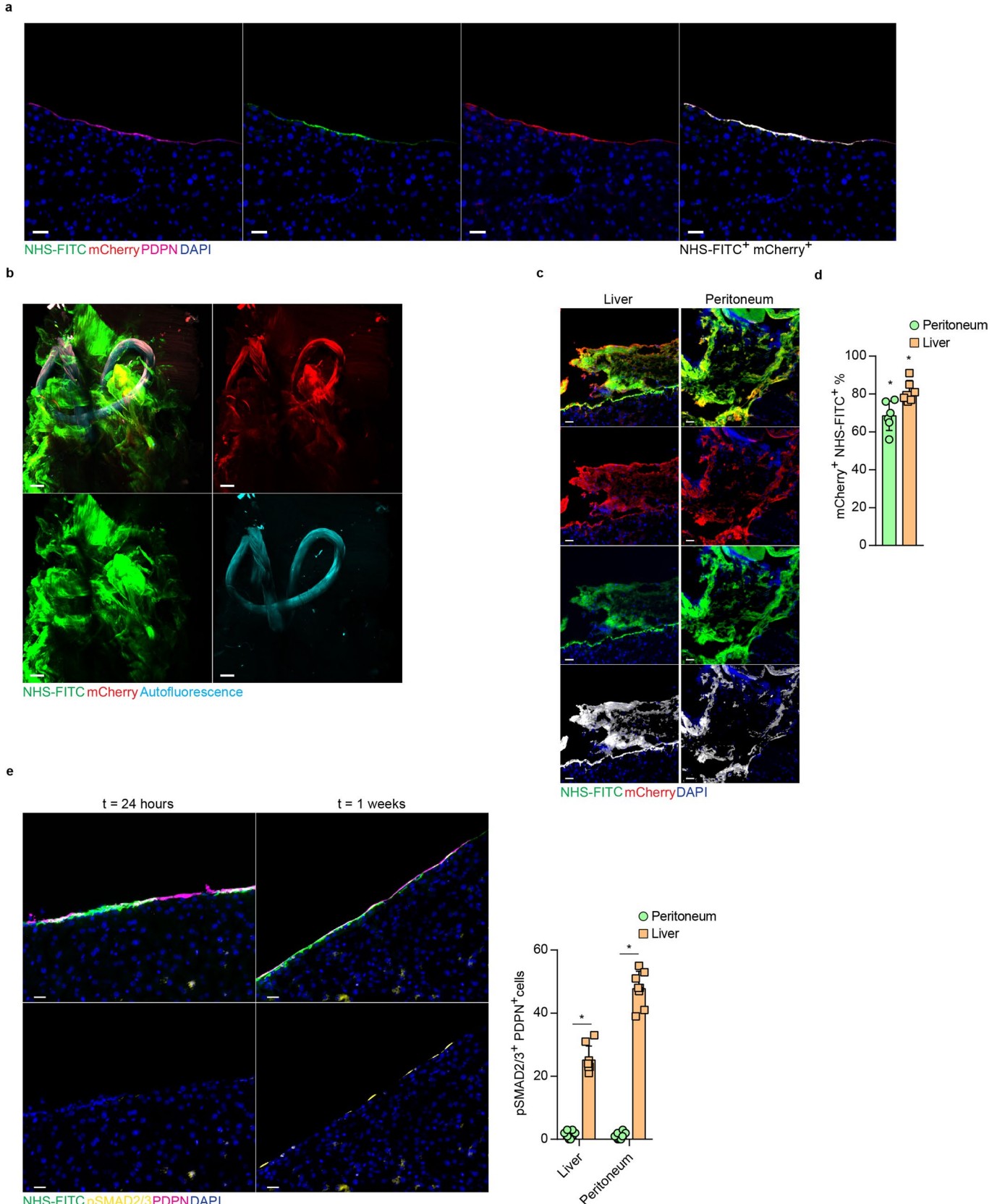

**a**

NHS-FITC mCherry PDPN DAPI

NHS-FITC⁺ mCherry⁺

**b**

NHS-FITC mCherry Autofluorescence

**c**

Liver    Peritoneum

NHS-FITC mCherry DAPI

**d**

mCherry⁺ NHS-FITC⁺ %

○ Peritoneum
□ Liver

**e**

t = 24 hours    t = 1 weeks

NHS-FITC pSMAD2/3 PDPN DAPI

pSMAD2/3⁺ PDPN⁺ cells

○ Peritoneum
□ Liver

Liver    Peritoneum

**Extended Data Fig. 4 | See next page for caption.**

**Extended Data Fig. 4 | Mesothelial cells are a source for transferred ECM. a**, Overview of CNA35 reporter system. Transduced mesothelial cells express CNA35 fused to mCherry, which binds to collagens. Representative immunostainings five days after intra peritoneal injection of CNA35 reporter system. Scale bar: 20 µm. n = six C57BL/6 J WT mice and three independent experiments. **b**, Representative light sheet images of laparotomy closure of animals transduced with CNA35-mcherry reporter and NHS-FITC surface label 24 hours post injury. n = six C57BL/6 J WT mice and three independent experiments. Scale bar: 500 µm. **c**, Representative histology images of animals transduced with CNA35-mcherry reporter and NHS-FITC surface label 24 hours post injury. n = six C57BL/6 J WT mice and three independent experiments. Scale bar: 50 µm. **d**, Quantification of FITC + CNA35 + double positive signal from c. n = six C57BL/6 J WT mice and three independent experiments. **e**, Representative immunohistological stainings of NHS-FITC labelled livers 24 hours and 1 week after injury. n = six C57BL/6 J WT mice and three independent experiments. Histology: 50 µm. Two-tailed Mann–Whitney; * P < 0.05.

**a**

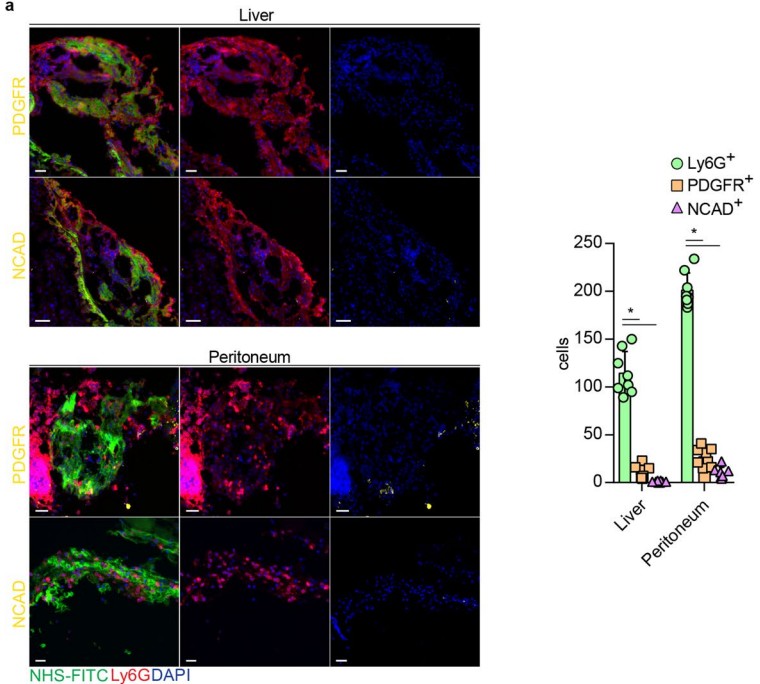

**b**

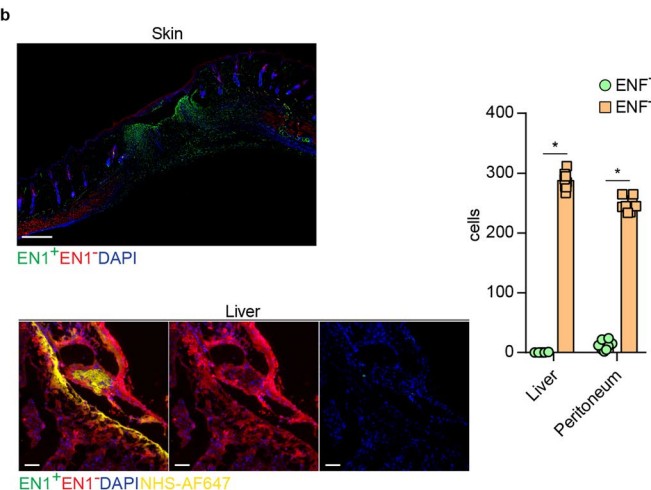

**c**

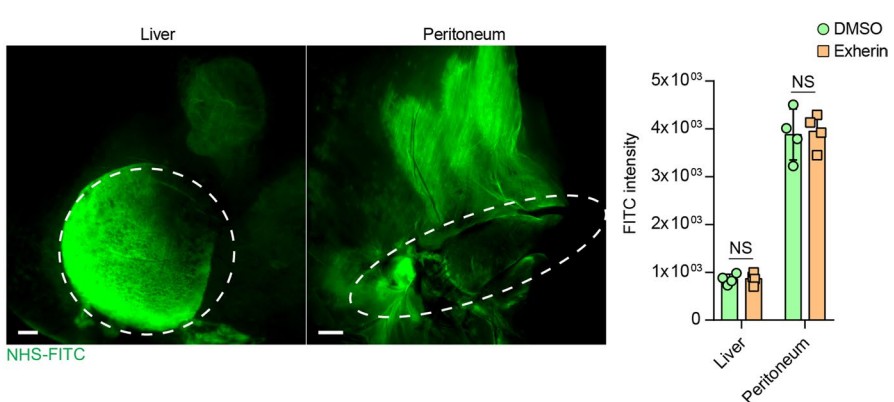

**Extended Data Fig. 5 | See next page for caption.**

**Extended Data Fig. 5 | Matrix movements on organ surfaces are independent of EN1 + fibroblasts. a**, Representative immunohistological stainings of organ wounds 24 hours after injury. Quantification of Ly6G+ neutrophils and PDGFR + NCad+ fibroblasts in organ wounds 24 hours post injury. n = six C57BL/6 J WT mice and three independent experiments. Scale bar: 50 μm. Two-tailed Mann–Whitney; * P < 0.05. **b**, Representative immunohistological stainings of EN1Cre;R26mTmG transgenic mouse organ wounds 24 hours after injury. Skin histology acts as EN1 reporter control, showing EN1[+] cells in skin wounds as previously reported[12]. n = six mice and four independent experiments. Scale bar: 50 μm. **c**, Representative images of organ surfaces in the presence of N-cadherin inhibitor Exherin post 24 hours after injury. n = four C57BL/6 J WT mice and three independent experiments. Two-tailed Mann–Whitney; NS = not significant. Scale bar: 500 μm.

Cecum

PBS    LPS

Peritoneum

PBS    LPS

Autofluorescence NHS-FITC

NHS-FITC

NHS-FITC DAPI Collagen1

NHS-FITC DAPI Collagen1

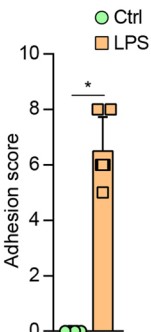

**Extended Data Fig. 6 | See next page for caption.**

**Extended Data Fig. 6 | Sterile inflammation activates matrix transfer and organ fibrosis.** Representative images of locally NHS labeled Organ surfaces followed by lipopolysaccharide injections five days after, samples were collected one week post LPS injection. n = six C57BL/6 J WT mice and four independent experiments. Stereomicroscope: Scale bar: 2000 μm. Histology: 30 μm. Adhesions were scored according to extended table 1. Two-tailed Mann–Whitney; * P < 0.05.

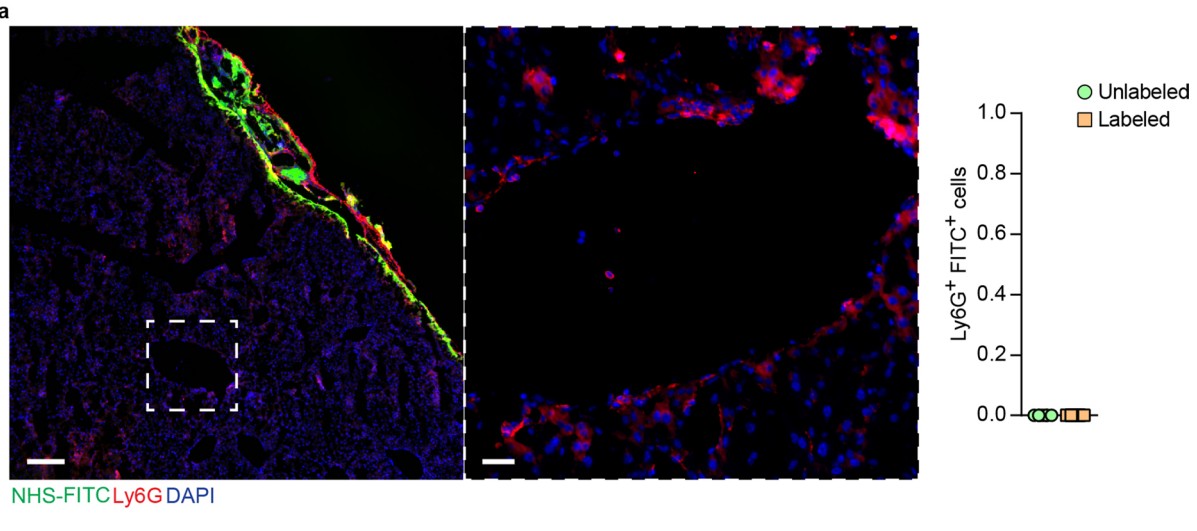

**Extended Data Fig. 7 | Neutrophils transport matrix directionally. a,** Representative images from livers surfaces with NHS-FITC 24 hours post injury. n = four biological replicates. Scale bar: Overview 500 μm; Histology: 30 μm. **b,** Flow cytometry analysis of blood from mice 24 hours after surgery and organ surface label. CD45+ and Ly6G+ were sorted for FITC+ signal, FITC labeled cells acted as positive control. n = five biological replicates and three independent experiments.

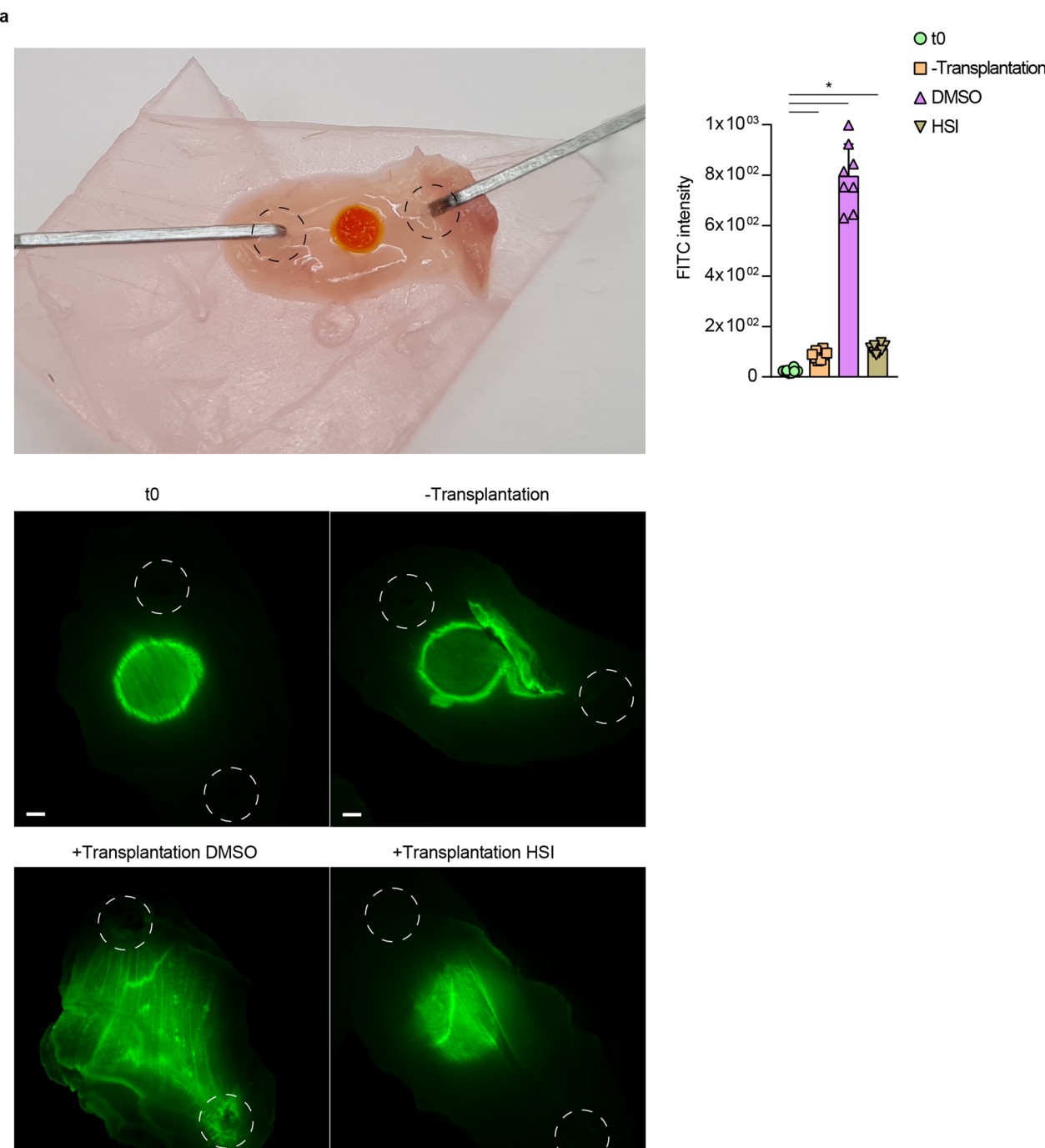

**Extended Data Fig. 8 | Neutrophils orchestrate matrix transfer via active HSF signaling *ex vivo*. a**, Representative images from cultivated NHS-FITC labelled peritoneal biopsy punches in the presence of neutrophils. Scale bar: 500 μm. n = five biological replicates and five independent experiments. Two-tailed Mann–Whitney; * P < 0.05; NS = not significant.

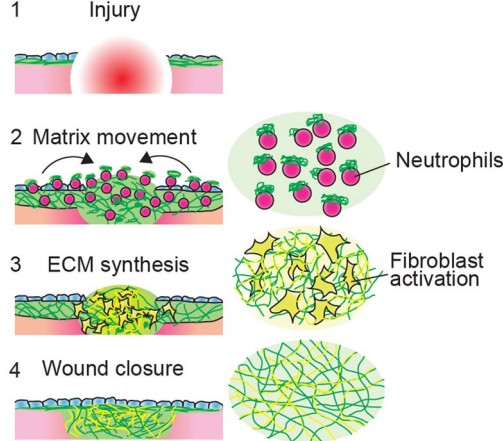

**Extended Data Fig. 9 | Model of organ surface wound healing. a**, Local wounding leads to recruitment of neutrophils carrying fluid matrix. This matrix provides a provisional wound closure, after a few days active fibroblasts provide a new deposition of extra cellular matrix, which is integrated into the fluid matrix.

# Reporting Summary

## Statistics

For all statistical analyses, confirm that the following items are present in the figure legend, table legend, main text, or Methods section.

| n/a | Confirmed | |
|---|---|---|
| ☐ | ☒ | The exact sample size (*n*) for each experimental group/condition, given as a discrete number and unit of measurement |
| ☐ | ☒ | A statement on whether measurements were taken from distinct samples or whether the same sample was measured repeatedly |
| ☐ | ☒ | The statistical test(s) used AND whether they are one- or two-sided<br>*Only common tests should be described solely by name; describe more complex techniques in the Methods section.* |
| ☒ | ☐ | A description of all covariates tested |
| ☒ | ☐ | A description of any assumptions or corrections, such as tests of normality and adjustment for multiple comparisons |
| ☐ | ☒ | A full description of the statistical parameters including central tendency (e.g. means) or other basic estimates (e.g. regression coefficient) AND variation (e.g. standard deviation) or associated estimates of uncertainty (e.g. confidence intervals) |
| ☐ | ☒ | For null hypothesis testing, the test statistic (e.g. *F*, *t*, *r*) with confidence intervals, effect sizes, degrees of freedom and *P* value noted<br>*Give P values as exact values whenever suitable.* |
| ☒ | ☐ | For Bayesian analysis, information on the choice of priors and Markov chain Monte Carlo settings |
| ☒ | ☐ | For hierarchical and complex designs, identification of the appropriate level for tests and full reporting of outcomes |
| ☒ | ☐ | Estimates of effect sizes (e.g. Cohen's *d*, Pearson's *r*), indicating how they were calculated |

*Our web collection on statistics for biologists contains articles on many of the points above.*

## Software and code

Policy information about availability of computer code

| Data collection | Leica Application Suite v4.8<br>AxioVision (Carl Zeiss) v4.8<br>ZEN2009 (Carl Zeiss)  v.5.5<br>Progenesis QIsoftware (version 4.1, Nonlinear Dynamics, Waters)<br>BD FACSDIVA v9.0 |
|---|---|
| Data analysis | ImageJ (1.52i)<br>Imaris (v9.1.0)<br>Mascot (Matrix Science, version 2.6.2)<br>GraphPad Prism (v.7.0, GraphPad)<br>flowjo 10.7. |

For manuscripts utilizing custom algorithms or software that are central to the research but not yet described in published literature, software must be made available to editors and reviewers. We strongly encourage code deposition in a community repository (e.g. GitHub). See the Nature Portfolio guidelines for submitting code & software for further information.

## Data

Policy information about availability of data

All manuscripts must include a data availability statement. This statement should provide the following information, where applicable:

- Accession codes, unique identifiers, or web links for publicly available datasets
- A description of any restrictions on data availability
- For clinical datasets or third party data, please ensure that the statement adheres to our policy

Data is in respective Source file.
The proteomics data are deposited at PRIDE under submission ID: px-submission #564638.
The scRNA-seq data are accessible to all. The raw files will be added to a public repository in the next 3-4 days.
Computer Code is available at: https://github.com/kadrisaf/Fluid_Matrix

# Field-specific reporting

Please select the one below that is the best fit for your research. If you are not sure, read the appropriate sections before making your selection.

☒ Life sciences    ☐ Behavioural & social sciences    ☐ Ecological, evolutionary & environmental sciences

For a reference copy of the document with all sections, see nature.com/documents/nr-reporting-summary-flat.pdf

# Life sciences study design

All studies must disclose on these points even when the disclosure is negative.

| Sample size | No statistical method was used to predetermine sample size. Required experimental sizes were estimated based on previous established protocols in the field (e.g. DOI: 10.1038/s41586-019-1794-y, ). The sample sizes were adequate as the differences between experimental groups were reproducible and were statistically significant, demonstrating the suitability of the sample size. All n values are clearly indicated within the figure legends. |
|---|---|
| Data exclusions | No data was excluded from the analysis. |
| Replication | All experiments were performed at least three times with similar results. |
| Randomization | Mice and ex vivo tissues were randomly divided into treatment groups. |
| Blinding | No experiments presented in this study required blinding. Critical in vivo adhesion scorings were on-off effects in all replicates. Samples were measured using automated methods across different assays with different personnel, which reduces the subjectivity to a justifiable minimum. |

# Reporting for specific materials, systems and methods

We require information from authors about some types of materials, experimental systems and methods used in many studies. Here, indicate whether each material, system or method listed is relevant to your study. If you are not sure if a list item applies to your research, read the appropriate section before selecting a response.

## Materials & experimental systems

| n/a | Involved in the study |
|---|---|
| ☐ | ☒ Antibodies |
| ☒ | ☐ Eukaryotic cell lines |
| ☒ | ☐ Palaeontology and archaeology |
| ☐ | ☒ Animals and other organisms |
| ☐ | ☒ Human research participants |
| ☒ | ☐ Clinical data |
| ☒ | ☐ Dual use research of concern |

## Methods

| n/a | Involved in the study |
|---|---|
| ☒ | ☐ ChIP-seq |
| ☐ | ☒ Flow cytometry |
| ☒ | ☐ MRI-based neuroimaging |

## Antibodies

| Antibodies used | Primary antibodies:<br>COLLAGEN I (Rockland, 600-401-103-0.1, 1:150)<br>COLLAGEN VI (Abcam, ab6588, 1:150)<br>Fibronectin (Abcam, ab23750, 1:100)<br>HSPG2 (Elabscience, E-AB-13507.20, 1:100), |
|---|---|

cleaved Caspase 3 (Abcam, 9661S, 1:100),
Laminin (Abcam, ab11575, 1:100),
PDPN (Abcam, ab11936, 1:100),
LY6G (Abcam, ab25377, 1:100),
F4/80 (Abcam, ab90247, 1:400),
TNF (Abcam, ab223352, 1:150),
ITGAM (Abcam, ab8878, 1:100),
ITGβ2 (Abcam, ab63388, 1:150),
CD45.1 (Abcam, ab23910, 1:150),
FPR1 (Abcam, ab113531, 1:100),
CD62L (Abcam, ab119834, 1:100),
YAP/TAZ, (Abcam, ab205270, 1:100),
PDGFRA (R&D systems, AF1062, 1:50) ,
phosphoSMAD2/3 (Abcam, 18338S, 1:100),
phosphoHSF (Elabscience, E-AB-20894.60, 1:100),
Ncadherin (Abcam, ab18203, 1:100),
GAPDH (Life Scientist, MA515738, 1:1000).
Kindlin3 (Sigma, PRS4797,1:1000)

FACS:
Reagent or Antibodies Cat.  Co.
APC anti-mouse CD45 , Rat IgG2b, kappa, Clone: 30-F11 100 µg 103112 Biolegend
Pacific Blue™ anti-mouse Ly-6G 100 µg 127612 Biolegend

Following products (1:500, Life technologies) against suitable species were used as secondary antibodies.
Alexa Fluor 488 Donkey Anti-Rabbit (A21206),
Alexa Fluor 594 Donkey Anti-Rabbit, (A21207),
Alexa Fluor 594 Donkey Anti-Goat, (A32758),
Alexa Fluor 647 Donkey Anti-Goat, (A-21447),
Alexa Fluor 647 Donkey Anti-Rabbit IgG, (A31573),
Alexa Fluor 647 Goat Anti-Rat, (A21247).

| Validation | All antibodies have been validated by the manufacturer and by multiple citations for reactivity against mouse or human. Abcam: "We use a variety of methods, including staining multi-normal human tissue microarrays (TMAs), multi-tumor human TMAs, and rat or mouse TMAs during antibody development. These high-throughput arrays allow us to check many tissues at the same time, providing uniformly as all tissues are exposed to the exact same conditions. "<br><br>Antibodies were additionally validated using respective isotype antibodies in immunofluorescence assays. Negative controls were performed in all staining procedures. |
| --- | --- |

# Animals and other organisms

Policy information about studies involving animals; ARRIVE guidelines recommended for reporting animal research

| Laboratory animals | Following mouse strains were used, both males and females, adult at 8-12 weels old: C57BL/6J wild type, B6.129P2-Lyz2tm1(cre)Ifo/J (Lyz2Cre), B6;129S6-Gt(ROSA)26Sortm14(CAG-tdTomato)Hze/J (Ai14), En1Cre (En1tm2(cre)Wrst), En1Cre (En1tm2(cre)Wrst) and R26mTmG (Gt(ROSA)26Sortm4(ACTB-tdTomato,-EGFP)Luo) mice were purchased from Jackson Laboratories or Charles River, R26CreER;floxKindlin3flox mice were a gift from R.F. and P.K.. All mice were bred and maintained in the Helmholtz Animal Facility in accordance with EU directive 2010/63. Animals were housed in individual ventilated cages and animal housing rooms were maintained at constant temperature at 20-24 °C and humidity at 45-65% with a 12 hours light cycle. Animals were supplied with water and chow ad libitum. All animal experiments were reviewed and approved by the Government of Upper Bavaria and conducted under strict governmental and international guidelines. This study is compliant with all relevant ethical regulations regarding animal research. |
| --- | --- |
| Wild animals | This study did not involve wild animals. |
| Field-collected samples | The study did not involve samples collected from field. |
| Ethics oversight | Government of Upper Bavaria, Germany. |

Note that full information on the approval of the study protocol must also be provided in the manuscript.

# Human research participants

Policy information about studies involving human research participants

| | |
|---|---|
| Population characteristics | Patients undergoing mechanic adhesiolysis between 50-65 years of age, both genders. |
| Recruitment | All human samples were obtained from surgery at the Department of Surgery, Klinikum rechts der Isar, Technical University of Munich. Adhesions were intraoperatively diagnosed and dissected from the respective organs and prepared for further analysis. |
| Ethics oversight | Approval of the local ethics committee of the Technical University of Munich, Germany (Nr. 173/18 S). |

Note that full information on the approval of the study protocol must also be provided in the manuscript.

# Flow Cytometry

## Plots

Confirm that:

☒ The axis labels state the marker and fluorochrome used (e.g. CD4-FITC).

☒ The axis scales are clearly visible. Include numbers along axes only for bottom left plot of group (a 'group' is an analysis of identical markers).

☒ All plots are contour plots with outliers or pseudocolor plots.

☒ A numerical value for number of cells or percentage (with statistics) is provided.

## Methodology

| | |
|---|---|
| Sample preparation | Murine blood were collected by cardiac puncture and diluted within PBS + heparin. Red blood cells were lysed twice with 1X BD Pharm lysing solution. Labelling was carried out in PBS+1%BSA. |
| Instrument | BD LSRII (4-laser, UV325, V405, B488, R640) |
| Software | BD FACSDIVA was used for acquisit and recording cells. Finally, gating, ploting were performed in flowjo 10.7. |
| Cell population abundance | Abundance of cell population gating were indicated figures. The purity of sorted cells were determined by flow cytometric analysis of sorted cells with the same gating strategy as during sorting. Cell populations were analyzed by gating CD45+Ly6G+ neutrophil. Depending on the experimental question, a fraction of FITC+ cells were calculated in neutrophil cluster. One of examples was plotted with numerical value of percentage and a ridgeline plot summarized FITC cell distribution in all groups. |
| Gating strategy | Intact cells were gated based FSC-A and SSC-A. We removed doublets by FSC-A and FSC-H, SSC-A and SSC-H. LIVE/DEAD fixable near-IR dead cell staining kit was used to exclude dead cells. Neutrophil was gated by CD45-APC and Ly6g-PB. |

☒ Tick this box to confirm that a figure exemplifying the gating strategy is provided in the Supplementary Information.

