## [Peer Review File · Nature Immunology]

Peer Review Information

Journal: Nature Immunology

Manuscript Title: Neutrophils direct preexisting matrix in to initiate repair of damaged organs

Corresponding author name(s): Yuval Rinkevich

Reviewer Comments & Decisions:

Decision Letter, initial version:
--

Subject: Decision on Nature Immunology submission NI-A32301

Message: 12th Jul 2021

Dear Dr. Rinkevich,

Your Article, "Neutrophils direct preexisting matrix in to initiate repair of damaged organs" has now been seen by 3 referees. While we find your work is of considerable potential interest, the reviewers have raised substantial concerns that must be addressed. As such, we cannot accept the current version of the manuscript for publication, but would be happy to consider a revised version that addresses these concerns, as long as novelty is not compromised in the interim.

At resubmission, please include a point-by-point "Response to referees" detailing how you have addressed each referee comment (please specify page and figure number). This response will be sent back to the referees along with the revised manuscript.

In addition, please include a revised version of any required reporting checklist. It will be available to referees (and, potentially, statisticians) to aid in their evaluation if the manuscript goes back for peer review. A revised checklist is essential for re-review of the paper.

The Reporting Summary can be found here:
<https://www.nature.com/documents/nr-reporting-summary.pdf>

When submitting the revised version of your manuscript, please pay close attention to our [href="https://www.nature.com/nature-research/editorial-policies/image-integrity">Digital Image Integrity Guidelines. and to the following points below:](https://www.nature.com/nature-research/editorial-policies/image-integrity)

- that unprocessed scans are clearly labelled and match the gels and western blots presented in figures.
- that control panels for gels and western blots are appropriately described as loading on sample processing controls

-- all images in the paper are checked for duplication of panels and for splicing of gel lanes.

You may use the link below to submit your revised manuscript and related files:
[redacted]

We hope to receive the revised manuscript within 6 months. If you cannot send it within this time, please let us know. We will be happy to consider your revision so long as nothing similar has been accepted for publication at Nature Immunology or published elsewhere.

Nature Immunology is committed to improving transparency in authorship. As part of our efforts in this direction, we are now requesting that all authors identified as 'corresponding author' on published papers create and link their Open Researcher and Contributor Identifier (ORCID) with their account on the Manuscript Tracking System (MTS), prior to acceptance. ORCID helps the scientific community achieve unambiguous attribution of all scholarly contributions. You can create and link your ORCID from the home page of the MTS by clicking on 'Modify my Springer Nature account'. For more information please visit www.springernature.com/orcid.

Thank you for the opportunity to review your work.

Sincerely,

Ioana Visan, Ph.D.
Senior Editor
Nature Immunology

Tel: 212-726-9207
Fax: 212-696-9752
www.nature.com/ni

Reviewers' Comments:

Reviewer #1:

Remarks to the Author:

This is a very exciting and interesting novel study showing that matrix during repair is not synthesized but rather transported from healthy sites. The neutrophils appear to be the transporters of this event. I only have some minor queries.

1) Can the authors block adhesion formation with Ly6G antibody? Do the neutrophils induce the adhesion formation or this a different mechanism from repair. This needs to be better elucidated.

2) How are the neutrophils getting to these injury sites. Are they arriving via the vasculature or via the peritoneum and then crawling to these sites? How do they end up in healthy sites? What recruits them there?

3) Blocking LFA-1 may block both crawling of neutrophils and their ability to transport matrix. Which role is LFA-1 playing. Is there a role for the other important CD18 integrin Mac-1 (CD11b/CD18).

4) What happens at the site where the collagen is removed? Are there holes or does the mesothelium quickly fill things in?

5) The myeloid cell reporter is not specific enough to conclude that only neutrophils transport matrix to the site of injury. Monocytes also infiltrate the site and could deliver matrix. Depleting neutrophils does not negate a potential role for monocytes since neutrophils could dig tunnels allowing monocytes to migrate into injury sites. Many other scenarios could be possible.

6) Many of the videos are quite difficult to follow. Arrows indicating where the authors would like the readership to look to demonstrate the point being made would be helpful.

Reviewer #2:

Remarks to the Author:

Fischer and colleagues describe a fascinating study regarding a novel paradigm in tissue repair. They provide evidence that matrix proteins from distant healthy tissue sites are moved toward an injury site. This early repair is dominantly mediated by such transfer rather than de novo synthesis by local fibroblasts. The authors provide additional evidence that neutrophils are the source of these 'moved' matrix proteins by bringing these proteins into the wounded area; a mechanism not dependent on early phagocytosis and release. In fact, the data suggest that the cells carry the proteins. Last but not least the data suggest that integrins and HSP signaling play a pivotal role opening up a novel route towards clinical intervention for deregulated scar formation. This is a very novel hypothesis with potentially very important consequence for our understanding of tissue repair. The article would, however, benefit when the following issues are carefully addressed (random order).

Major:

1. The authors have not paid any attention to the fact that multiple studies in the last decade have provided compelling evidence that the neutrophil compartment is very

heterogeneous. These studies show that different functionalities exhibited by neutrophils are confined to certain phenotypes (see for an excellent review: Silvestre-Roig C et al. Trends Immunol. 2019 Jul;40(7):565-583). It is important to understand which neutrophil phenotype is at work in moving the matrix. Or are all neutrophil phenotypes involved in this process.

2. The order of events is a bit difficult to comprehend. The prevailing consensus in the neutrophil field is that these cells extravasate to the tissue in response to inflammatory cues which is mediated by control both at the level of expression (e.g. selectins) as well as inside-out control (e.g. integrins). So how do the authors envision how a resting neutrophil 'knows' to extravasate in healthy tissue in response to a distant wound, reverse migrate back to the vasculature taking the matrix cargo with it, and finds the wound later on. Do the authors now suggest that the complete neutrophil compartment is taking cargo in response to a small wound? This seems of course unlikely. So please share with the readers of NI what the hypothesis/ideas around this conundrum is/are.

3. The focus on Kindlin-3 is smart as this protein is essential for inside-out control of all integrin families (Beta-1, beta-2, beta-3 and beta-7). The clinical phenotype of LADIII (human kindling-3 deficiency) should be carefully discussed in support of the current study mainly carried out with murine cells (see also below).

4. A seemingly obvious experiment is the determination of the amount of fluorescent cargo and the number of cells carrying cargo by flowcytometry. A supported concept is that neutrophils can reverse migrate from the tissues to the blood and find their way to wounds such as suggested here. One might expect fluorescent neutrophils in the blood during transfer of the cells from healthy tissues towards the wound. So please provide such obvious experiment or explain why this experiment is doomed to fail.

5. The human experiments mentioned in lines 184-188 are far from convincing. Consider removing or better experiments should be performed.

6. The LTB4 experiments and their interpretation are very confusing. It suggests that LTB4-induced swarming is essential. The 5-lipoxygenase pathway has been targeted with several inhibitors in human trials in the past including FLAP inhibitors (important protein in the leukotriene pathway). As far as I am aware these trials did not show deregulated tissue repair. This issue should be carefully addressed. Also neutrophils are notoriously sensitive for aspecific effects of pharmacological inhibitors. Try not to be too convincing in the conclusion (eg. HSP inhibitors) of experiments manipulating neutrophils with such inhibitors.

7. In conclusion, the authors make an important point that their strategy with fluorescent tags such as a nontoxic N-hydroxysuccinimide ester fluorescein (NHS-FITC) are not toxic, which is fundamental of their approach. This might be the case but the real question is whether this strategy is inert: with other words is the tagging not leading to activation of local neutrophils? This is important as these local activated neutrophils might be primed to "find" damaged tissue spots. This would be a trivial explanation. A similar mechanism is seen during the role of neutrophils in the modulation of the pre-metastatic niche.

Minor:

8. Figure 1. It is surprising that the authors choose FITC for their initial experiments as this fluorescent molecule is highly sensitive for photo bleaching. How did the authors deal with bleaching when re-visiting stained areas.

9. Figure 1. Have the authors ruled the possibility that the FITC label detaches from the stained matrix protein and aspecifically stained proteins at a distance?

10. Can the authors rule out that the neutrophil is producing matrix proteins as suggested

in the literature (Bastian et al. Neutrophils contribute to fracture healing by synthesizing fibronectin+ extracellular matrix rapidly after injury. Clin Immunol. 2016 Mar;164:78-84). The argument of the non-canonical amino acids might not be attributable for neutrophils, which might have synthesized and stored these proteins early in differentiation such as the granule proteins. Mature end-stage neutrophils have a limited potency to translate proteins.

11. Line 248: there is no consensus in the neutrophil field that neutrophils use proteases to transmigrate in such a way that endothelium is damaged and leaky. Please be more careful. Two articles from medium ranked journals is not sufficient support.

12. Extended figure 7: TNF, FPR1 en L-selectin are not routinely used activation markers for neutrophils. CD11b, CD66B, CD63 are more generally used. The issue around L-selectin is more of a problem as this protein is shed from the surface upon activation, and high expression of L-selectin is a marker of non-activated rather than activated cells.

13. The issue of lack of longevity of neutrophils should be addressed. The traditional view is that neutrophils have a very short life-span (7-9 hrs) leading to an estimated production of these cells of 10E11 cells per day. Does this fit with the neutrophil involved in the model of tissue repair? Or do the authors support a longer lifespan of neutrophils in their models? Please discuss.

Reviewer #3:

Remarks to the Author:

This is a comprehensive and technically excellent study that provides molecular evidence for neutrophils transferring matrix across to sites of injury at the surface of organs such as the liver and peritoneum. The authors clearly demonstrate a molecular mechanism for this phenomenon which involves 'piggy-back' carrying of pre-existing matrix that requires a combination of heat shock-integrin-kindlin signaling. The importance of this property of neutrophils was confirmed by the authors demonstrating that the neutrophil-transferred matrix is required for forming a wound-healing foundation and formation of long-lasting scars by the subsequent activities of fibroblasts. Most impressively pharmacological inhibition of heat shock signaling prevented matrix transfer and development of adhesions. The study is a technical tour de force and is illustrated with beautiful images and videos that are a delight to observe. However, there are limitations to the study that temper enthusiasm for the study appealing to the general wound healing field and in addition there are some technical matters and issues relating to the manuscript and its organisation that should be considered.

Main conceptual issues:

1. The experiments and findings are certainly fascinating but appear to be restricted to wounding by electroporation at the surface of organs involving matrix transfer in the mesothelium. What is the relevance of neutrophil-mediated matrix transfer in the context of the internal structures of organs and physiological/environmental rather than surgical injuries? For example, if the liver is acutely injured with an hepatotoxin it is known that neutrophils are rapidly recruited to sites of injury and well ahead of fibroblast activation. Yet it appears from the literature that neutrophils are not essential for this type of hepatic wound-repair or indeed necessarily for the development of fibrosis. It would be important to ask if neutrophil transfer of matrix occurs in these types of injury models and if so then does it have a physiological role which if perturbed modulates wound repair and/or fibrosis. Further, what about other organs not addressed by the authors such as the lung

or kidney?

2. What is the origin of the neutrophils in the injury models used by the authors, are they cells that are recruited from the periphery to the tissue in response to damage or are they patrolling neutrophils? What is the developmental profile of the neutrophils that carry matrix? Are they mature or immature cells, do they have an activated inflammatory phenotype and are these features required for matrix transport? Can the scRNAseq data (which needs better explanation) be used to interrogate these questions in detail followed by phenotype manipulation experiments.

3. What are the neutrophil chemoattractant signals involved in recruitment to the wound? In the experiments Lipoxin A4 is used, however the authors do not explain how neutrophils swarm to the site of injury or what the major attractors are.

4. Following transport of matrix to the site of injury how is it subsequently released from the neutrophil and how is it subsequently deposited and built into the local granulation tissue? What happens to the neutrophil once transfer is complete?

Answering these questions would certainly provide greater clarity of the generality of the phenomenon identified by the authors and extend the mechanistic reach of the work.

Technical and Manuscript Organisation Issues:

1. The text and figures are in places difficult to follow which can make what is a beautiful study a rather taxing read. The figures are also organised and annotated in a complicated and confusing way, certainly the C/C' type nomenclature is not at all helpful and it is often difficult to match experimental outlines to images and to data in graphs. The authors are urged to make the figures much easier for the reader to follow and understand.

2. Fig 1 This figure jumps all over the place, it needs to be more logically organised for the reader to follow. How specific is NHS-FITC for matrix lining organ surface? Does it label the surface proteins on cells present at the surface? Fig1d – this needs a T0 to make check specificity of labelling - 30min data could just be to less intense diffuse staining on edges. Fig1e(&ext 1b) - does intensity of signal change after injury? Overall signal should be the same, but brightness should be less? Fig1f&g - was this corrected for tissue weight? How was this standardised between mice. Could mice be monitored longitudinally using a non-destructive method such as intravital imaging?

3. Exten Fig 1- useful to show Caspase3+ or some other measure of cell death/damage after injury. What is the depth of injury and subsequent transfer of labelled surface matrix?

4. Exten Fig 2- is this the same for the liver? There should be many PDGFR+ cells in the liver under normal uninjured conditions, although perhaps not at the mesothelium which again brings into question the broader relevance of the findings for wound healing?

5. Figure 3. Panel A is confusing/misleading - need better schematic to understand exactly what has been done. The schematic suggests that labelling & injury were performed on the same organ - or is peritoneum labelled and liver/caecum injured as in figure 2. It would be useful to know the kinetics/temporal transfer of specific matrix. Panel C- no error bars? What is the distribution of matrix under homeostatic conditions (ie prior to injury T0). Need to know this before can make conclusions in D as these components might already be increased under normal conditions.

6. Exten fig 3. More information is needed for the postsurgical adhesions. Sites of adhesions? What organs? Site of injury? Liver? Caecum? How long after surgery were samples collected? Same timescale as mice or longer? No Extended Fig. 3c - line 200 should be 4c?

7. Exten fig5 – no control provided to show that report actually works.

8. Exten fig 6. For completeness and given the use of lyz2 mouse there is a need to more completely characterise monocyte/macrophage infiltration into the site of injury and find a better, preferably a genetic way to rule out these cells as contributing to wound repair. Clodronate is often not that effective and removes only a subset of macrophages. There should also be a control for the reporter
9. Exten fig 8. Important data - should possibly come sooner? The authors show depletion of neutrophils using ly6g but there are no macrophage counts following clodronate treatment to show that it has worked, as discussed above it is not considered the ideal way to rule out monocyte/macrophage involvement. Does neutrophil depletion alter wound severity? This is important to determine.
10. Exten Fig8e – is this truly “swarming” or just neutrophil chemotaxis in general? What about other neutrophil recruitment factors that are known to be involved in wound repair, it would be interesting to determine a hierarchy for this response using CXCR2 inhibitors, anti-IL6, FPR1KO mice etc.
11. Exten Fig 9 e&f- a more comprehensive profiling of neutrophil gene expression is required before and after injury needed - 2 genes are not sufficient to draw conclusions that neutrophil gene expression changes significantly. Are there 2 populations of neutrophils in 7-day IRE - high and low CD11b?
12. The single cell RNAseq data require more explanation, exactly how was this achieved and controlled? Neutrophils are notoriously difficult to produce this type of data from due to their low abundance of mRNA and especially if numbers of cells isolated are only small. Where exactly in the tissue were the cells isolated from, how were they proven to be pure neutrophils and how many cells were used to generate the data sets? The data should also be very extensive enabling a more detailed description of the phenotype of neutrophils in these models.
13. Exten fig11. CD11b/CD18 expression sustained up to day 7? Panel C- more description required in the text - what organ was injured? Using what injury? Line 283 - 12b should be 11b. Need an experiment putting neutrophils back in after injury is established not at the same time?
14. Fig 4. Labelling (a,a' etc) is very confusing. Panel B - what about neutrophils from non-wound site of injured mice? Panel E - does HSI alter healing time? No good having an absence of adhesions if the primary wound doesn't heal properly.

Author Rebuttal to Initial comments

Reviewers' Comments:

Reviewer #1:

Remarks to the Author:

This is a very exciting and interesting novel study showing that matrix during repair is not synthesized but rather transported from healthy sites. The neutrophils appear to be the transporters of this event. I only have some minor queries.

1)Can the authors block adhesion formation with Ly6G antibody? Do the neutrophils induce the adhesion formation or this a different mechanism from repair. This needs to be better elucidated.

A: We and others have previously shown that adhesion formation is reduced significantly using the monoclonal anti Ly6G antibody (Gr-1 (1A8)), please see: <https://ashpublications.org/bloodadvances/article/3/18/2713/374948/Neutrophil-and-monocyte-kinetics-play-critical>. The new mechanism of matrix transfer we are describing in this paper occurs through heat shock-Integrin-Kindlin3 signaling. a sentence will be added to the discussion section to discuss the novelty of our new mechanistic findings over previous neutrophil publications in adhesion formation.

2) How are the neutrophils getting to these injury sites. Are they arriving via the vasculature or via the peritoneum and then crawling to these sites? How do they end up in healthy sites? What recruits them there?

A: As we do not detect neutrophils with NHS-FITC labeled matrix within the vessels (Extended Fig. 9a), we believe matrix transport occurs through neutrophil crawling along serous membranes (organ surfaces and cavities) where they are profusely detected carrying labeled proteins.

To cement the histological findings, we have now added an additional experiment in which we examined circulating neutrophils in the blood by flow cytometry analysis that includes neutrophil markers and with anti FITC antibodies. We could not detect significant amounts of neutrophils with FITC+ matrix cargo in the blood of animals after surgery. This further indicates that neutrophils carrying matrix do not extravasate, but remain within organs where we detect crawling on surfaces with transported matrix towards wounds (Extended Fig. 9b).

3) Blocking LFA-1 may block both crawling of neutrophils and their ability to transport matrix. Which role is LFA-1 playing. Is there a role for the other important CD18 integrin Mac-1 (CD11b/CD18).

A: In our integrin blocking experiments with anti CD11b and anti CD18, we find that neutrophils are still present within wounds but are all devoid of FITC+ matrix (please see Extended Fig. 14). Our analysis of these wounds shows that general neutrophil recruitment is not blocked as cells are still physically present within these wounds, yet matrix transport is specifically inhibited (Extended Fig. 14b). We conclude from these experiments that only matrix transfer, not crawling, is impaired by our anti CD11b/CD18 blocking regimen.

4) What happens at the site where the collagen is removed? Are there holes or does the mesothelium quickly fill things in?

A: We thank the reviewer for raising the question of remote tissue effects to a local injury. We have now examined the labeling sites, remote to the wound, which have been labeled with NHS-ester and across different time points post injury. We have performed histology and immunostaining for active TGFb signaling in mesothelial cells. We detect significant amounts of pSMAD2/3 positive mesothelial cells one-week post-surgery and away from the wound sites. This suggests a remote mechanism where several days post matrix transport, mesothelial cells slowly initiate de novo matrix deposition to renew

these matrix depleted areas. This new dataset has now been integrated into the revised manuscript (please see Extended Fig. 4e and line 216-218).

5) The myeloid cell reporter is not specific enough to conclude that only neutrophils transport matrix to the site of injury. Monocytes also infiltrate the site and could deliver matrix. Depleting neutrophils does not negate a potential role for monocytes since neutrophils could dig tunnels allowing monocytes to migrate into injury sites. Many other scenarios could be possible.

A: Our neutrophil transplantation experiments, as well as macrophage depletion experiments with Clodronate liposomes, gives us reduced macrophage numbers but with no effect on matrix transport itself, leading us to consider that neutrophils are the primary transporters of matrix, at least in this specific injury model. We agree that other injury models such as sterile inflammation could activate additional cellular mechanisms potentially including matrix transport by macrophages, but which go beyond the specific scope of our mechanistic findings on neutrophils. We therefore have now more carefully stated the role of macrophages in matrix transport specifically in our wound repair model (please see line 263-264).

6) Many of the videos are quite difficult to follow. Arrows indicating where the authors would like the readership to look to demonstrate the point being made would be helpful.

A: We have considered adding arrows into the videos, and have decided this might add confusion to understanding the events unfolding in our fast-moving videos. However, we have taken the reviewer's comment to heart and have now added arrows to plot the distance traveled by neutrophils with matrix cargo. Extended Fig. 7a shows highlights of the example from the videos.

Remarks to the Author:

Fischer and colleagues describe a fascinating study regarding a novel paradigm in tissue repair. They provide evidence that matrix proteins from distant healthy tissue sites are moved toward an injury site. This early repair is dominantly mediated by such transfer rather than de novo synthesis by local fibroblasts. The authors provide additional evidence that neutrophils are the source of these ‘moved’ matrix proteins by bringing these proteins into the wounded area; a mechanism not dependent on early phagocytosis and release. In fact, the data suggest that the cells carry the proteins. Last but not least the data suggest that integrins and HSP signaling play a pivotal role opening up a novel route towards clinical intervention for deregulated scar formation. This is a very novel hypothesis with potentially very important consequence for our understanding of tissue repair. The article would, however, benefit when the following issues are carefully addressed (random order).

A: We thank the reviewer for their suggestions and have now carefully addressed these comments by incorporating into the revised manuscript new experiments, new analysis and additional textual changes that address their concerns. The changes and additions are individually detailed in our point-by-point response below.

Major:

1. The authors have not paid any attention to the fact that multiple studies in the last decade have provided compelling evidence that the neutrophil compartment is very heterogeneous. These studies show that different functionalities exhibited by neutrophils are confined to certain phenotypes (please see for an excellent review: Silvestre-Roig C et al. Trends Immunol. 2019 Jul;40(7):565-583). It is important to understand which neutrophil phenotype is at work in moving the matrix. Or are all neutrophil phenotypes involved in this process.

A: We agree with reviewer that the question of which subpopulation of neutrophils is responsible for matrix transport is a very exciting one. We have now incorporated a complete re-analysis of our Single Cell RNA-Seq data implementing state-of-the-art RNA velocity, fate probability mapping and hierarchical clustering (please see extended Fig. 13) to look even more closely into neutrophil subsets in our datasets.

This new dataset shows that wound neutrophils in our models display substantial cellular heterogeneity between time points, in particular in the later injury response (Extended Fig. 13b). In addition, we now show that neutrophil subsets that are transcriptionally mature and differentiated (Fig. 13c) gradually increase in numbers over time, as indicated by increased activation of apoptotic factors and expression of age-related genes (please see line 297-305).

We further deep-dive into these diverse neutrophil subsets by immunostaining for suggested neutrophil subset markers (Figure 4, Extended Fig. 7, 8, 10, 14 and 15). Because we see broad expression of markers in neutrophils, we hypothesize that a variety of phenotypes are involved in this process (please see line 417-435).

2. The order of events is a bit difficult to comprehend. The prevailing consensus in the neutrophil field is that these cells extravasate to the tissue in response to inflammatory cues which is mediated by control both at the level of expression (e.g. selectins) as well as inside-out control (e.g. integrins). So how do the authors envision how a resting neutrophil 'knows' to extravasate in healthy tissue in response to a distant wound, reverse migrate back to the vasculature taking the matrix cargo with it, and finds the wound later on. Do the authors now suggest that the complete neutrophil compartment is taking cargo in response to a small wound? This seems of course unlikely. So please share with the readers of NI what the hypothesis/ideas around this conundrum is/are.

A: Our data indicates that local injury leads to organ-wide recruitments of matrix by neutrophils, rather than from a single location. We also show now that circulating neutrophils lack matrix cargo, and therefore matrix transport occurs exclusively within the interstitial spaces of organs. We therefore believe that matrix transport occurs during this specific stage of contact between neutrophils and the organ's ECM as they relocate towards the wound site. We do not see any evidence of matrix cargo on circulating neutrophils, indicating that any return back to the blood occurs in the absence of matrix cargo (please see extended Figure 9a and b) . We have expanded our Discussion section to better place matrix cargo dynamics with neutrophil dynamics (please see line 417-435).

3. The focus on Kindlin-3 is smart as this protein is essential for inside-out control of all integrin families (Beta-1, beta-2, beta-3 and beta-7). The clinical phenotype of LADIII (human kindling-3 deficiency) should be carefully discussed in support of the current study mainly carried out with murine cells (please see also below).

A: We thank the reviewer and have now included a segment in the Discussion section where we address LADIII (please see line 410-415).

4. A seemingly obvious experiment is the determination of the amount of fluorescent cargo and the number of cells carrying cargo by flowcytometry. A supported concept is that neutrophils can reverse migrate from the tissues to the blood and find their way to wounds such as suggested here. One might expect fluorescent neutrophils in the blood during transfer of the cells from healthy tissues towards the wound. So please provide such obvious experiment or explain why this experiment is doomed to fail.

A: We have now performed the suggested experiment by looking closely into the possibility that circulating neutrophils carry matrix cargo. We do not observe any neutrophils carrying matrix within the blood. This is supported by high resolution immunohistochemical stains and by flow cytometry analysis of circulating neutrophils. Please see Extended Fig. 9. The Discussion section has now included a new segment that better explains the dynamics of matrix cargo with neutrophils all based on these new findings (please see line 271-274 and 417-435).

5. The human experiments mentioned in lines 184-188 are far from convincing. Consider removing or better experiments should be performed.

A: We have now changed the text in the revised manuscript to more cautiously interpret our human data (please see line 196-198). We believe that immunolabeling of markers of matrix cargo on human adhesion samples does indicate that human fibrous adhesions consist of the same transported ECM material as in mouse adhesions, and suggesting that matrix transport occurs in human adhesions as well.

6. The LTB4 experiments and their interpretation are very confusing. It suggests that LTB4-induced swarming is essential. The 5-lipoxygenase pathway has been targeted with several inhibitors in human trials in the past including FLAP inhibitors (important protein in the leukotriene pathway). As far as I am aware these trials did not show deregulated tissue repair. This issue should be carefully addressed. Also neutrophils are notoriously sensitive for aspecific effects of pharmacological inhibitors. Try not to be too convincing in the conclusion (eg. HSP inhibitors) of experiments manipulating neutrophils with such inhibitors.

A: We thank the reviewer for this comment and have now reworded our statements on the potential role of LTB4 in swarming in the revised manuscript (please see line 280-284).

7. In conclusion, the authors make an important point that their strategy with fluorescent tags such as a nontoxic N-hydroxysuccinimide ester fluorescein (NHS-FITC) are not toxic, which is fundamental of their approach. This might be the case but the real question is whether this strategy is inert: with other words is the tagging not leading to activation of local neutrophils? This is important as these local activated neutrophils might be primed to find damaged tissue spots. This would be a trivial explanation. A similar mechanism is seen during the role of neutrophils in the modulation of the pre-metastatic niche.

A: We apologize for not inserting this essential control in the draft do to space limitations. However, these essential controls have been performed and are now included in Extended Fig. 1c. NHS ester labeling of unwounded tissues compared to labelled tissues does not lead to inflammation or to neutrophil recruitments (Extended Fig. 1c.). Furthermore, our transplantation experiments, i.e. when new neutrophils are externally added, leads to matrix transport, which cannot be explained by local neutrophils alone as matrix transport is absent in all our control experiments. To further prove the specificity of matrix transport to injury, we have independently of NHS esters, performed genetic tracing experiments with Col1-Flag reporter, showing same matrix movement dynamics occurs genetically with Col1-Flag and in the absence of NHS labeling. Thus, we show in 2 independent approaches (chemical using NHS-ester and genetic) as well as in control experiments that matrix transport occurs in response to injury alone, and is absent from non-injured tissues.

Minor:

8. Figure 1. It is surprising that the authors choose FITC for their initial experiments as this fluorescent molecule is highly sensitive for photo bleaching. How did the authors deal with bleaching when re-visiting stained areas.

A: Matrix transport is also observed with additional fluorescent dyes, including AF594, Pacific Blue and AF647 (please see Figure 2c-g, extended Fig. 1d, extended Fig. 3). Using microscope setups with short fluorescence imaging durations enable minimal bleaching in our experiments using NHS-FITC. Furthermore, all microscopy tissue samples and protein lysates were kept in cooled, light protected vials in the dark. We have now added these technical setups to our Methods section (please see line 662-663 and 684-685).

9. Figure 1. Have the authors ruled the possibility that the FITC label detaches from the stained matrix protein and aspecifically stained proteins at a distance?

A: Yes we have controlled for this possibility in all our experiments. Once NHS esters are applied, the molecules are covalently bound to amines and are incapable of detaching from the labeled proteins (DOI: 10.1016/0003-2697(90)90267-d and 10.1007/BF00986726). We have further confirmed this point in our control experiments, both *in vivo* and *ex vivo*, where we see absence of detachment of NHS-esters. Please see Fig 1b and Extended Fig 6a showing absence of leakage/detachment.

10. Can the authors rule out that the neutrophil is producing matrix proteins as suggested in the literature (Bastian et al. Neutrophils contribute to fracture healing by synthesizing fibronectin+ extracellular matrix rapidly after injury. Clin Immunol. 2016 Mar;164:78-84). The argument of the non-canonical amino acids might not be attributable for neutrophils, which might have synthesized and stored these proteins early in differentiation such as the granule proteins. Mature end-stage neutrophils have a limited potency to translate proteins.

A: We show that matrix transport occurs in the absence of *de novo* synthesis. Both in chemical experiments where matrix transport is unaffected in the presence of collagen biosynthesis inhibitors in extended fig. 10a, as well as in our neutrophil transplantation experiments where NHS-FITC positive matrix is labeled prior to administering neutrophils (Fig. 4c and extended Fig. 14c). since we do not label neutrophils with our method, the labeled matrix found in wounds matrix is derived remotely from transported material. These findings suggest that the matrix present in the wound was transported and not synthesized *de novo*.

11. Line 248: there is no consensus in the neutrophil field that neutrophils use proteases to transmigrate in such a way that endothelium is damaged and leaky. Please be more careful. Two articles from medium ranked journals is not sufficient support.

A: We thank the reviewer and have reworded our statements (please see line 265).

12. Extended figure 7: TNF, FPR1 en L-selectin are not routinely used activation markers for neutrophils. CD11b, CD66B, CD63 are more generally used. The issue around L-selectin is more of a problem as this protein is shed from the surface upon activation, and high expression of L-selectin is a marker of non-activated rather than activated cells.

A: As suggested by the reviewer, we removed L-Selectin and included new panels of immunostaining with CD66b and CD63 (please see Extended Fig 7c).

13. The issue of lack of longevity of neutrophils should be addressed. The traditional view is that neutrophils have a very short life-span (7-9 hrs) leading to an estimated production of these cells of 10^{11} cells per day. Does this fit with the neutrophil involved in the model of tissue repair? Or do the authors support a longer lifespan of neutrophils in their models? Please discuss.

A: We feel this issue of cellular longevity of neutrophils is an important topic but one which is beyond the scope of the paper. However, we have tried to address this issue by re-examining our scRNAseq datasets for apoptosis and age-related genes (please see Extended Fig. 13c.) We can now show that neutrophils upregulate diverse apoptosis genes already at 24 hours post injury. Our time-resolved measurements of matrix transport also shows that transport dynamics rapidly flattens after an initial sharp increase. We therefore speculate that neutrophil half-life as well as uptake/recruitments of new neutrophils into sites of wounds, both shape the extent and dynamics of matrix transport into wounds. A sentence has been added to the Discussion section to highlight both neutrophil half-life and recruitments in shaping the overall dynamics of matrix transport into wounds (please see line 424-427)

Reviewer #3:

Remarks to the Author:

This is a comprehensive and technically excellent study that provides molecular evidence for neutrophils transferring matrix across to sites of injury at the surface of organs such as the liver and peritoneum. The authors clearly demonstrate a molecular mechanism for this phenomenon which involves 'piggy-back' carrying of pre-existing matrix that requires a combination of heat shock-integrin-kindlin signaling. The importance of this property of neutrophils was confirmed by the authors demonstrating that the neutrophil-transferred matrix is required for forming a wound-healing foundation and formation of long-lasting scars by the subsequent activities of fibroblasts. Most impressively pharmacological inhibition of heat shock signaling prevented matrix transfer and development of adhesions. The study is a technical tour de force and is illustrated with beautiful images and videos that are a delight to observe. However, there are limitations to the study that temper enthusiasm for the study appealing to the general wound healing field and in addition there are some technical matters and issues relating to the manuscript and its organisation that should be considered.

A: We thank the reviewer for the enthusiastic comments on the comprehensiveness and technical quality of the work.

Main conceptual issues:

1. The experiments and findings are certainly fascinating but appear to be restricted to wounding by electroporation at the surface of organs involving matrix transfer in the mesothelium. What is the

relevance of neutrophil-mediated matrix transfer in the context of the internal structures of organs and physiological/environmental rather than surgical injuries? For example, if the liver is acutely injured with an hepatotoxin it is known that neutrophils are rapidly recruited to sites of injury and well ahead of fibroblast activation. Yet it appears from the literature that neutrophils are not essential for this type of hepatic wound-repair or indeed necessarily for the development of fibrosis. It would be important to ask if neutrophil transfer of matrix occurs in these types of injury models and if so then does it have a physiological role which if perturbed modulates wound repair and/or fibrosis. Further, what about other organs not addressed by the authors such as the lung or kidney?

A: We and others have previously shown that adhesion formation is reduced significantly using neutrophil targeted monoclonal anti Ly6G antibody (Gr-1 (1A8)), please see: <https://ashpublications.org/bloodadvances/article/3/18/2713/374948/Neutrophil-and-monocyte-kinetics-play-critical>. We therefore believe matrix transport by neutrophils is a very important mechanism for injuries to organ surfaces. We do agree that if the question is how universal is matrix transport to all injury models such as, for example, sterile inflammation models or other forms of liver injury using hepatotoxins, where the trigger might be different, then they should be studied on a case by case basis.

Our study primarily focuses on lesions to organ surfaces but we have also included additional systemic injury models such as LPS, revealing commonality of matrix transport in both systemic inflammation and acute injury models. Overall, the injury models we used are not limited to electroporation, but include 8 different, clinically relevant injury models. These are:

1. Liver surface injury via electroporation
2. surgical laparotomy to the abdominal wall.
3. cecal brushing
4. peritoneal brushing
5. chemical irritation via Talcum administration to the peritoneum
6. chemical irritation to the cecum
7. LPS administration into the abdomen
8. generation of hypoxic tissue pockets to the peritoneum

There are many potential additional models of injury that could be included. However, these experiments would need to include new animal protocol approvals (currently approval time estimated to take between 6-8 months) and we feel is beyond the scope of this already breaching animal study.

2. What is the origin of the neutrophils in the injury models used by the authors, are they cells that are recruited from the periphery to the tissue in response to damage or are they patrolling neutrophils? What is the developmental profile of the neutrophils that carry matrix? Are they mature or immature cells, do they have an activated inflammatory phenotype and are these features required for matrix

transport? Can the scRNAseq data (which needs better explanation) be used to interrogate these questions in detail followed by phenotype manipulation experiments.

A: We thank the reviewer for their many suggestions and questions. Clearly these findings open up a range of fascinating additional questions that are beyond the scope of a single study which is already beaming with animal models and experimentation.

To address the reviewer's point regarding scRNAseq methodology: We have now expanded the description of our single cell RNA seq Methodology. In brief, three livers were taken per experimental group and were pooled for each sequencing run. For each liver, the electroporated area was punched out with a circular 4 mm biopsy punch, and subsequently minced with fine scissors into small pieces (approximately 1 mm²). The equivalent, but non-injured area was used in control livers. The resulting fragments were enzymatically digested and a total of 250,000 cells were loaded for DropSeq at a final concentration of 100 cells/μL (see line 724-750) .

We have also addressed the important points raised by this reviewer with regards to neutrophil profiling in this revision step: We have now re-analyzed the scRNAseq datasets to include a more focused analysis of neutrophils, their subsets, and differentiation stage (please see Extended Fig. 13) focusing on the neutrophil profile. By analyzing markers of differentiated neutrophil subsets, we find that matrix transport is mediated by various heterogeneous neutrophils. In other words, this appears to be a central conserved process across neutrophil differentiated states. To experimentally confirm these findings, we have now performed chemotaxis manipulation experiments in vivo using CXCR2 and CXCR4 inhibitors. Both have been implied in neutrophil chemotaxis and increased CXCR4 expression is one of our identified neutrophil differentiation markers seven days post IRE (please see Extended Fig. 13d). We find that CXCR2 inhibition significantly blocked neutrophil recruitment and matrix transport, whereas CXCR4 inhibition showed no significant effect (please see Extended Fig. 10b). This demonstrates that while CXCR2 seems to play an essential role in the immediate matrix transfer as chemoattractant, CXCR4 function seems not to be direct and more later down in the repair process.

3. What are the neutrophil chemoattractant signals involved in recruitment to the wound? In the experiments Lipoxin A4 is used, however the authors do not explain how neutrophils swarm to the site of injury or what the major attractors are.

A: As suggested by the reviewer, we have now performed additional experiments with the neutrophil chemoattractant signals activating CXCR2 and CXCR4 (please see Extended Fig. 10b). These 2 new in vivo experiments, in addition to our existing NOS synthase inhibitory experiments, support a role for these 2 separate chemoattractant signals in matrix transport. Neutrophils infiltrating wounds are known to have active iNOS and we are able to prevent neutrophil recruitment and matrix transport into wounds, with iNOS inhibition alone as well as with CXCR2. Based on our new chemotaxis experiments, we conclude that CXCR2 ligands and reactive oxygen species are the 2 major attractants for matrix transfer by neutrophils. The new dataset and text have been included into the revised manuscript (please see line 280-284).

4. Following transport of matrix to the site of injury how is it subsequently released from the neutrophil and how is it subsequently deposited and built into the local granulation tissue? What happens to the neutrophil once transfer is complete? Answering these questions would certainly provide greater clarity of the generality of the phenomenon identified by the authors and extend the mechanistic reach of the work.

A: At the moment we can only speculate about this deposition process. However, our scRNAseq analysis suggests that the neutrophils differentiate/age on site or go through apoptosis after arriving and depositing matrix. The transported matrix would then remain and be modified by other mediators and incorporated into the wound. In our figure nr 2 we show that the moving protein elements are cross-linked after being transferred by neutrophils, supporting this hypothesis.

Technical and Manuscript Organisation Issues:

1. The text and figures are in places difficult to follow which can make what is a beautiful study a rather taxing read. The figures are also organised and annotated in a complicated and confusing way, certainly the C/C’ type nomenclature is not at all helpful and it is often difficult to match experimental outlines to images and to data in graphs. The authors are urged to make the figures much easier for the reader to follow and understand.

A: We have carefully modified and improved the organization and clarity of our figures in the revised manuscript.

2. Fig 1 This figure jumps all over the place, it needs to be more logically organised for the reader to follow. How specific is NHS-FITC for matrix lining organ surface? Does it label the surface proteins on cells present at the surface? Fig1d; this needs a T0 to make check specificity of labelling - 30min data could just be to less intense diffuse staining on edges. Fig1e- does intensity of signal change after injury? Overall signal should be the same, but brightness should be less? Gig1f&g - was this corrected for tissue weight? How was this standardised between mice. Could mice be monitored longitudinally using a non-destructive method such as intravital imaging?

A: Our matrix tracing protocol does not mark mesothelial cells. Please see example in (Extended Fig. 1a). We have inserted the control images for figure 1d as requested by the reviewer.

With regards to the question on fluorescence quantifications, we have now expanded our Methodology section to include a more detailed description of the fluorescence quantification methods. In brief, tissues were collected with a 2mm biopsy punch, lysed and exact same protein amounts were determined via BCA assay and FITC signal was determined via fluororeader.

With regards to intravital imaging, we believe this requires developing a completely new setup and animal protocol approval that, with current restrictions, would significantly delay any publication by an additional 8 months and would go beyond the already breaching scope of this study.

3. Exten Fig 1- useful to show Caspase3+ or some other measure of cell death/damage after injury. What is the depth of injury and subsequent transfer of labelled surface matrix?

A: We now include immunolabeling images as well as quantification measurements of cell death across control experiments and with NHS-ester labeled organ surfaces (please see Extended Fig. 1b, 1b'). Injuries to liver surfaces such as liver electroporation, brushing of the peritoneum or cecum led to matrix transport laterally. Whereas a more deep transection wound to the peritoneum resulted in both caspase3 activation in deeper tissues as well as matrix transport into deep peritoneal tissue with subsequent wound closure (please see Figure 1 c and extended Fig. 1c and Extended Fig 2b).

4. Exten Fig 2- is this the same for the liver? There should be many PDGFR+ cells in the liver under normal uninjured conditions, although perhaps not at the mesothelium which again brings into question the broader relevance of the findings for wound healing?

A: We now include immunolabeling of fibroblasts with antibodies against PDGFR. We have quantified PDGFR+ fibroblastic cells in homeostatic non-injured conditions as well three days and two weeks post injury (please see Extended Fig. 2d and e). We also show a broad relevance of matrix transport in more deep tissue injuries. We indeed see extensive transport of labeled ECM into deep tissues when a deep peritoneal injury is inflicted (please see Extended Fig. 2b).

5. Figure 3. Panel A is confusing/misleading - need better schematic to understand exactly what has been done. The schematic suggests that labelling & injury were performed on the same organ - or is peritoneum labelled and liver/caecum injured as in figure 2. It would be useful to know the kinetics/temporal transfer of specific matrix. Panel C- no error bars? What is the distribution of matrix under homeostatic conditions (ie prior to injury T0). Need to know this before can make conclusions in D as these components might already be increased under normal conditions.

A: We have now improved our schematic, integrated error bars and statistical tests (Figure 3).

In brief, animals were subjected to local abdominal labeling and local distant injury, local injury and wound closure. 24 hours post injury samples (original label and wound site) were collected using biopsy punches. Tissue lysis was performed according to established methods and same quantities of protein lysate was subjected to pulldown, elution and to mass spectrometry analysis.

We included now a new graph in figure 3c that illustrates the total matrix pools of each organs. In fact, the matrix pools of the organs are different, so the peritoneum, which has more collagens, can transport more collagens to wounds.

This suggests that this is also an explanation for why we see the fluid matrix of the peritoneum on livers in postoperative adhesions as seen in Figure 2f.

We have also amended the text to better explain that each individual mice had a single organ that was subjected to NHS-ester labeling followed by injury (only one injury model per mouse). Kinetics of matrix transfer is clearly indicated in Figure 1f and 1g (please see line 164-169)

We repeatedly see absence of matrix distribution in control, non-injured mice. This is reflected in Figure 1b and Extended Fig. 6

6. Exten fig 3. More information is needed for the postsurgical adhesions. Sites of adhesions? What organs? Site of injury? Liver? Caecum? How long after surgery were samples collected? Same timescale as mice or longer? No Extended Fig. 3c - line 200 should be 4c?

A: We have now revised the figure legend and text to include more detail into the labeling Methodology (please see line 600-604). The surgical adhesions are generated between peritoneum and cecum by injuring both sides and collecting samples after 4 week post injury. The age of the human adhesion samples is unknown. This is because the exact adhesion formation in humans cannot be observed or the diagnosis can only be made when the abdomen is opened. All of the human adhesions were formed between the peritoneum and intestines.

7. Exten fig5 – no control provided to show that report actually works.

A: We have now included skin samples of the EN1^{Cre}R26^{mTmG} double transgenic reporter mouse showing clear En1-positive cells in skin wounds (Extended Fig. 5b).

8. Exten fig 6. For completeness and given the use of lyz2 mouse there is a need to more completely characterise monocyte/macrophage infiltration into the site of injury and find a better, preferably a genetic way to rule out these cells as contributing to wound repair. Clodronate is often not that effective and removes only a subset of macrophages. There should also be a control for the reporter

A: We have now quantified macrophage numbers in our Clodronate experiments and show that Clodronate significantly reduces macrophage numbers, but has no significant effect on the amount of matrix transported (Extended Fig. 8a).

9. Exten fig 8. Important data - should possibly come sooner? The authors show depletion of neutrophils using ly6g but there are no macrophage counts following clodronate treatment to show that it has worked, as discussed above it is not considered the ideal way to rule out monocyte/macrophage involvement. Does neutrophil depletion alter wound severity? This is important to determine.

A: We now include quantifications of macrophages from Clodronate administration experiments, as well as a better description of wound outcomes in vivo in our neutrophil depletion experiments (please see line).

10. Exten Fig8e – is this truly “swarming” or just neutrophil chemotaxis in general? What about other neutrophil recruitment factors that are known to be involved in wound

repair, it would be interesting to determine a hierarchy for this response using CXCR2 inhibitors, anti-IL6, FPR1KO mice etc.

A: We have now performed additional sets of in vivo experiments with CXCR2 and CXCR4 chemical blockers to monitor the effects of these 2 chemoattractant pathways on matrix transport. We show that matrix transport is dependent on CXCR2, but not CXCR4 chemotaxis (please see Extended Fig. 10b).

11. Exten Fig 9 e&f- a more comprehensive profiling of neutrophil gene expression is required before and after injury needed - 2 genes are not sufficient to draw conclusions that neutrophil gene expression changes significantly. Are there 2 populations of neutrophils in 7-day IRE - high and low CD11b?

A: We have now included a new figure that addresses the reviewer's point on neutrophil gene expression. By re-analyzing our scRNAseq data we find that several neutrophil cell clusters contain cells with low and high ITGAM/CD11b (please see PBP-Fig. 1) seven days post IRE ITGAM. The distributions of low/highITGAM/CD11b appears universal across all neutrophil clusters:

PBP-Figure 1: Itgam has no influence on clustering of neutrophils seven days post IRE. Left) High resolution cluster of neutrophils seven days post IRE identifies 8 clusters. Right) Subclustered neutrophils show either high or low Itgam expression.

12. The single cell RNAseq data require more explanation, exactly how was this achieved and controlled? Neutrophils are notoriously difficult to produce this type of data from due to their low abundance of mRNA and especially if numbers of cells isolated are only small. Where exactly in the tissue were the cells isolated from, how were they proven to be pure neutrophils and how many cells were used to generate the data sets? The data should also be very extensive enabling a more detailed description of the phenotype of neutrophils in these models.

A: Per this reviewer's request we have now expanded on the Methods description of our neutrophil purification and scRNAseq. in brief, three livers per experimental group were pooled for each sequencing run. For each liver, the electroporated area was punched out with a circular 4 mm biopsy punch, and subsequently minced with fine scissors into small pieces (approximately 1 mm²). The equivalent, but non-injured area was used in control livers. The resulting tissue fragments were enzymatically digested and a total of 250,000 cells were loaded for DropSeq at a final concentration of 100 cells/ μ L (please see line 724-750).

As represented in extended figure 11a we detected 1565 neutrophils. We used Ly6G, MMP8, MMP9 and S100a8 expression as criteria (please see Extended figure 11d). In addition, We now also include a more in depth description of neutrophil subsets and their gene expression profiles based on this reviewer's request (please see extended Fig. 13d)

13. Exten fig11. CD11b/CD18 expression sustained up to day 7? Panel C- more description required in the text - what organ was injured? Using what injury? Line 283 - 12b should be 11b. Need an experiment putting neutrophils back in after injury is established not at the same time?

A: The injured organs have now been highlighted and the organ names inserted above the images. Since we are interested in the transport after 24 hours, we did not examine the expression of CD11b and CD18 after one week. The reason for introducing neutrophils early, and not at a late stage of injury, is because we observe matrix movement by neutrophils very early on (within minutes/hours) after organ injury, and therefore we aimed to target this early dynamics of matrix transfer into wounds in our rescue experiments.

14. Fig 4. Labelling (a, etc) is very confusing. Panel B - what about neutrophils from non-wound site of injured mice? Panel E - does HSI alter healing time? No good having an absence of adhesions if the primary wound doesn't heal properly.

A: In our HIS experiment the wounds are closed, and macroscopically there are no apparent healing alterations. Mice from these experiments show no abnormal or significant weight loss or signs of morbidity. We have added a sentence to the text in order to better explain our HSI phenotypes (please see line 371).

Decision Letter, first revision:

Subject: Decision on Nature Immunology submission NI-A32301A

Message: Dear Dr. Rinkevich,

Thank you for your response to the reviewers' comments on your manuscript "Neutrophils direct preexisting matrix in to initiate repair of damaged organs". We are happy to inform you that if you revise your manuscript appropriately in response to the referees' comments and our editorial requirements your manuscript should be publishable in Nature Immunology.

Please revise your manuscript according with the reviewers' comments and as outlined in your letter. At resubmission, please include a point-by-point response to the referees' comments, noting the pages and lines where the changes can be found in the revision. Please highlight the changes in the revised manuscript as well.

We are trying to improve the quality and transparency of methods and statistics reporting in our papers (please see our editorial in the May 2013 issue). Please update the Life Sciences Reporting Summary, and supplements if applicable, with any information relevant to any new experiments and upload it (as a Related Manuscript File) along with the files for your revision. If nothing in the checklist has changed, please upload the current version again.

TRANSPARENT PEER REVIEW

Nature Immunology offers a transparent peer review option for new original research manuscripts submitted from 1st December 2019. We encourage increased transparency in peer review by publishing the reviewer comments, author rebuttal letters and editorial decision letters if the authors agree. Such peer review material is made available as a supplementary peer review file. **Please state in the cover letter 'I wish to participate in transparent peer review' if you want to opt in, or 'I do not wish to participate in transparent peer review' if you don't.** Failure to state your preference will result in delays in accepting your manuscript for publication.

ORCID

Nature Immunology is committed to improving transparency in authorship. As part of our efforts in this direction, we are now requesting that all authors identified as 'corresponding author' on published papers create and link their Open Researcher and Contributor Identifier (ORCID) with their account on the Manuscript Tracking System (MTS), prior to acceptance. ORCID helps the scientific community achieve unambiguous attribution of all scholarly contributions. For more information please visit www.springernature.com/orcid.

Before resubmitting the final version of the manuscript, if you are listed as a corresponding author on the manuscript, please follow the steps below to link your account on our MTS with your ORCID. If you don't have an ORCID yet, you will be able to create one in minutes. If you are not listed as a corresponding author, please ensure that the corresponding author(s) comply.

1. From the home page of the [MTS](https://mts-ni.nature.com/cgi-bin/main.plex) click on '**Modify my Springer Nature account**' under '**General tasks**'.
2. In the '**Personal profile**' tab, click on '**ORCID Create/link an Open Researcher**

Contributor ID(ORCID). This will re-direct you to the ORCID website.

3a. If you already have an ORCID account, enter your ORCID email and password and click on '**Authorize**' to link your ORCID with your account on the MTS.

3b. If you don't yet have an ORCID, you can easily create one by providing the required information and then click on '**Authorize**'. This will link your newly created ORCID with your account on the MTS.

IMPORTANT: All authors identified as 'corresponding authors' on the manuscript must follow these instructions. Non-corresponding authors do not have to link their ORCIDs, but please note that it will not be possible to add/modify ORCIDs at proof. Thus, if they wish to have their ORCID added to the paper, they must also follow the above procedure prior to acceptance.

To support ORCID's aims, we only allow a single ORCID identifier to be attached to one account. If you have any issues attaching an ORCID identifier to your Manuscript Tracking System account, please contact the [Platform Support Helpdesk](http://platformsupport.nature.com/).

We hope that you will support this initiative and supply the required information. Should you have any query or comments, please contact immunology@us.nature.com.

Nature Immunology has now transitioned to a unified Rights Collection system which will allow our Author Services team to quickly and easily collect the rights and permissions required to publish your work. Once your paper is accepted, you will receive an email in approximately 10 business days providing you with a link to complete the grant of rights. If you choose to publish Open Access, our Author Services team will also be in touch at that time regarding any additional information that may be required to arrange payment for your article.

For information regarding our different publishing models please see our [Transformational Journals](https://www.springernature.com/gp/open-research/transformational-journals) page. If you have any questions about costs, Open Access requirements, or our legal forms, please contact ASJournals@springernature.com.

In recognition of the time and expertise our reviewers provide to Nature Immunology's editorial process, we would like to formally acknowledge their contribution to the external peer review of your manuscript entitled "Neutrophils direct preexisting matrix in to initiate repair of damaged organs". For those reviewers who give their assent, we will be publishing their names alongside the published article.

When you are ready to submit your revised manuscript, please use the URL below to submit the revised version: [redacted]

We hope to receive your revised manuscript in 10 days, by 3rd Jan 2022. Please let us know if circumstances will delay submission beyond this time. If you have any questions please do not hesitate to contact me.

Sincerely,

Ioana Visan, Ph.D.
Senior Editor
Nature Immunology

Tel: 212-726-9207
Fax: 212-696-9752
www.nature.com/ni

Reviewer #1 (Remarks to the Author):

Fantastic paper and wonderful rebuttal.

Accepted

Reviewer #2 (Remarks to the Author):

The manuscript has improved considerably. A few last but important lingering issues remain that must be dealt with.

The order of events is a bit difficult to comprehend. The prevailing consensus in the neutrophil field is that these cells extravasate to the tissue in response to inflammatory cues which is mediated by control both at the level of expression (e.g. selectins) as well as inside-out control (e.g. integrins). So how do the authors envision how a resting neutrophil 'knows' to extravasate in healthy tissue in response to a distant wound, reverse migrate back to the vasculature taking the matrix cargo with it, and finds the wound later on. Do the authors now suggest that the complete neutrophil compartment is taking cargo in response to a small wound? This seems of course unlikely. So please share with the readers of NI what the hypothesis/ideas around this conundrum is/are.

A: Our data indicates that local injury leads to organ-wide recruitments of matrix by neutrophils, rather than from a single location. We also show now that circulating neutrophils lack matrix cargo, and therefore matrix transport occurs exclusively within the interstitial spaces of organs. We therefore believe that matrix transport occurs during this specific stage of contact between neutrophils and the organ's ECM as they relocate towards the wound site. We do not see any evidence of matrix cargo on circulating neutrophils, indicating that any return back to the blood occurs in the absence of matrix cargo (please see extended Figure 9a and b) . We have expanded our Discussion section to better place matrix cargo dynamics with neutrophil dynamics (please see line 417-435)

1. I am very confused by the answer. The question was how a blood normal neutrophil "knows" (which mechanisms?) to extravasate into a healthy tissue spot to collect matrix before returning to the vasculature and bring the matrix to the injured site. The new information that no blood neutrophils carry any cargo adds to the confusion. The authors do not explain it nor put forward a hypothesis supported by data. Also the new text (lines 417-435) do not shed light on this conundrum. This issue must be dealt with.

2. Actually the new data on CD18 block showing no phenotype on neutrophil distribution (only effect on cargo) also adds to the confusion. LAD1 (CD18 deficiency) in humans is characterized by neutrophilia in blood and absence of neutrophils in (diseased) tissues.

These issues need to be clarified by testable hypotheses and not by speculation that basically describes what is seen.

3. Also the answer to question 12 is concerning. Now a relevant but difficult to explain true finding (high expression of CD62L) is simply deleted. Please show the data and explain rather than delete.

4. I do not agree that matrix cargo dynamics is now better placed with neutrophil kinetics. Now the issues above become even more complex. The authors now claim that the pseudotime analysis suggests that different phenotypes found in the tissue is explained by new cells going to the tissues as the half-life is less than a day and changing neutrophil phenotypes occur up to 7 days. The data in the literature (Lahoz-Beneytez et al. Blood) on a short half-life assumes that neutrophil half-life is best described by an ODE (stochastic) model without any supporting data that this is true. Longevity of neutrophils in the tissue is a perfect alternative hypothesis. Please comment or explain this view is wrong.

Reviewer #3 (Remarks to the Author):

The authors have adequately addressed the points that I raised and are to be congratulated on an outstanding study.

Author Rebuttal, first revision:

Reviewer #1

(Remarks to the Author)

Fantastic paper and wonderful rebuttal.

Accepted

Authors: We thank the reviewer for this feedback and for their acceptance of our manuscript.

Reviewer #2

(Remarks to the Author)

The manuscript has improved considerably. A few last but important lingering issues remain that must be dealt with.

The order of events is a bit difficult to comprehend. The prevailing consensus in the neutrophil field is that these cells extravasate to the tissue in response to inflammatory cues which is mediated by control both at the level of expression (e.g. selectins) as well as inside-out control (e.g. integrins). So how do the authors envision how a resting neutrophil 'knows' to extravasate in healthy tissue in response to a distant wound, reverse migrate back to the vasculature taking the matrix cargo with it, and finds the wound later on. Do the authors now suggest that the complete neutrophil compartment is taking cargo in response to a small wound? This seems of course unlikely. So please share with the readers of NI what the hypothesis/ideas around this conundrum is/are.

Authors: We thank the reviewer for their comments here. To clarify, as the reviewer rightly states, the consensus of the field is that neutrophils and other inflammatory cells know to extravasate and enter injured tissues due to a plethora of different chemical and physical signals such as inflammatory cues. Our manuscript has not been designed to specifically address the reason for extravasation, as there is a wealth of pre-existing knowledge here, but rather what events happen post-extravasation. To clarify, with respect to the reviewer's comment "...reverse migrate back to the vasculature taking the matrix cargo with it, and finds the wound later on", this is not the phenomenon we see. Our data indicates that neutrophils extravasate in response to the microenvironmental cues alluded to above, and at this point "pick up" cargo within the interstitial place and transport it within the tissue towards the wound site. We do not have evidence suggesting that neutrophils which have cargo on board leave the tissue and re-enter the vasculature.

Reviewer: Do the authors now suggest that the complete neutrophil compartment is taking cargo in response to a small wound? This seems of course unlikely.

Author: This is a very interesting question, but beyond the scope of the experimental design utilised in this study. Here pan-neutrophils were tracked and monitored for their collection of cargo and translocation of that material during the course of injury and wound repair response within the tissue. We agree with the reviewer that a subselection of neutrophil populations may be interesting to evaluate and ascertain their individual propensity towards cargo movement in the wound healing response. We would not like to speculate on what these data would demonstrate at this point, however.

A: Our data indicates that local injury leads to organ-wide recruitments of matrix by neutrophils, rather than from a single location. We also show now that circulating neutrophils lack matrix cargo, and therefore matrix transport occurs exclusively within the interstitial spaces of organs. We therefore believe that matrix transport occurs during this specific stage of contact between neutrophils and the organ's ECM as they relocate towards the wound site. We do not see any evidence of matrix cargo on circulating neutrophils, indicating that any return back to the blood occurs in the absence of matrix cargo (please see extended Figure 9a and b). We have expanded our Discussion section to better place matrix cargo dynamics with neutrophil dynamics (please see line 417-435)

1. I am very confused by the answer. The question was how a blood normal neutrophil "knows" (which mechanisms?) to extravasate into a healthy tissue spot to collect matrix before returning to the vasculature and bring the matrix to the injured site. The new information that no blood neutrophils carry any cargo adds to the confusion. The authors do not explain it nor put forward a hypothesis supported by

data. Also the new text (lines 417-435) do not shed light on this conundrum. This issue must be dealt with.

Authors: Please see above comments. To reiterate, we do not see movement back into the circulation of neutrophils with cargo, rather movement of those cells through the interstitial space within the tissue towards the site of injury ie a movement of matrix through the tissue towards the injury site, where that matrix accumulates.

2. Actually the new data on CD18 block showing no phenotype on neutrophil distribution (only effect on cargo) also adds to the confusion. LAD1 (CD18 deficiency) in humans is characterized by neutrophilia in blood and absence of neutrophils in (diseased) tissues. These issues need to be clarified by testable hypotheses and not by speculation that basically describes what is seen.

Authors: The reviewer raises an important point with respect to the impact of systemic CD18 deficiency and its impact on the blood and tissues. It was for this reason that the outlined experiments were designed to specifically exert a CD18 blockade in a local, controlled area of the wound site. Our purpose was not to evaluate the effect of CD18 blockade on extravasation, there is already significant knowledge in this area, but rather the role of this integrin on neutrophils in cargo pick up and translocation of that cargo within the wound area. To address this the experimental design involved specific placement of neutrophils directly into the tissues to see, if once there, they could pick up the cargo and secondly whether they could then move that cargo towards the injury site. Here our data demonstrated that CD18 is directly implicated in the pick up and movement of cargo within the interstitial space.

3. Also the answer to question 12 is concerning. Now a relevant but difficult to explain true finding (high expression of CD62L) is simply deleted. Please show the data and explain rather than delete.

Authors: Our re-analysis of single cell RNAseq showed a heterogeneous mixture of neutrophils in the wound with L-selectin being expressed by many neutrophils. We are happy to add these data back into the manuscript (See Extended Figure 7).

4. I do not agree that matrix cargo dynamics is now better placed with neutrophil kinetics. Now the issues above become even more complex. The authors now claim that the pseudotime analysis suggests that different phenotypes found in the tissue is explained by new cells going to the tissues as the half-life is less than a day and changing neutrophil phenotypes occur up to 7 days. The data in the literature (Lahoz-Beneytez et al. Blood) on a short half-life assumes that neutrophil half-life is best described by an ODE (stochastic) model without any supporting data that this is true. Longevity of neutrophils in the tissue is a perfect alternative hypothesis. Please comment or explain this view is wrong.

Authors: We agree with the reviewer that this is an interesting phenomenon and one that will require further study. Our data does not address the aspect of longevity of neutrophil life upon extravasation. The experiments have demonstrated that there is a constant flow of neutrophils from the vasculature into the interstitial space, and upon extravasation those neutrophils can exert different phenotypes and perform roles integral to the wound healing response, such as cargo translocation. The data does not indicate that the same neutrophils persist for 7 days, but rather there is a continuous presence of neutrophils, through this perpetual flow from the vasculature, into the tissue to perform and maintain their healing response.

Reviewer #3

(Remarks to the Author)

The authors have adequately addressed the points that I raised and are to be congratulated on an outstanding study. Authors: We thank the reviewer for this feedback

Decision Letter, second revision:

Subject: Your manuscript, NI-A32301B

Message: Our ref: NI-A32301B

31st Jan 2022

Dear Dr. Rinkevich,

Thank you for your patience as we've prepared the guidelines for final submission of your Nature Immunology manuscript, "Neutrophils direct preexisting matrix in to initiate repair of damaged organs" (NI-A32301B). Please carefully follow the step-by-step instructions provided in the attached file, and add a response in each row of the table to indicate the changes that you have made. Please also check and comment on any additional marked-up edits we have proposed within the text. Ensuring that each point is addressed will help to ensure that your revised manuscript can be swiftly handed over to our production team.

When you upload your final materials, please include a point-by-point response to any remaining reviewer comments and please make sure to upload your checklist.

In recognition of the time and expertise our reviewers provide to Nature Immunology's editorial process, we would like to formally acknowledge their contribution to the external

peer review of your manuscript entitled "Neutrophils direct preexisting matrix in to initiate repair of damaged organs". For those reviewers who give their assent, we will be publishing their names alongside the published article.

Nature Immunology offers a Transparent Peer Review option for new original research manuscripts submitted after December 1st, 2019. As part of this initiative, we encourage our authors to support increased transparency into the peer review process by agreeing to have the reviewer comments, author rebuttal letters, and editorial decision letters published as a Supplementary item. When you submit your final files please clearly state in your cover letter whether or not you would like to participate in this initiative. Please note that failure to state your preference will result in delays in accepting your manuscript for publication.

Cover suggestions

As you prepare your final files we encourage you to consider whether you have any images or illustrations that may be appropriate for use on the cover of Nature Immunology.

Nature Immunology has now transitioned to a unified Rights Collection system which will allow our Author Services team to quickly and easily collect the rights and permissions required to publish your work. Approximately 10 days after your paper is formally accepted, you will receive an email in providing you with a link to complete the grant of rights. If your paper is eligible for Open Access, our Author Services team will also be in touch regarding any additional information that may be required to arrange payment for your article.

Please note that *Nature Immunology* is a Transformative Journal (TJ). Authors may publish their research with us through the traditional subscription access route or make their paper immediately open access through payment of an article-processing charge (APC). Authors will not be required to make a final decision about access to their article until it has been accepted. [29](https://www.springernature.com/gp/open-Find out more about Transformational Journals.

If you have any questions about costs, Open Access requirements, or our legal forms, please contact ASJournals@springernature.com.

Authors may need to take specific actions to achieve compliance with funder and institutional open access mandates. For submissions from January 2021, if your research is supported by a funder that requires immediate open access (e.g. according to Plan S principles) then you should select the gold OA route, and we will direct you to the compliant route where possible. For authors selecting the subscription publication route our standard licensing terms will need to be accepted, including our self-archiving policies. Those standard licensing terms will supersede any other terms that the author or any third party may assert apply to any version of the manuscript.

Please use the following link for uploading these materials: [redacted]

Best regards,

Elle Morris
Senior Editorial Assistant
Nature Immunology
Phone: 212 726 9207
Fax: 212 696 9752
E-mail: immunology@us.nature.com

On behalf of

Ioana Visan, Ph.D.
Senior Editor
Nature Immunology

Tel: 212-726-9207
Fax: 212-696-9752
www.nature.com/ni

Final Decision Letter:

Subject: Decision on Nature Immunology submission NI-A32301C

Message: In reply please quote: NI-A32301C

Dear Dr. Rinkevich,

I am delighted to accept your manuscript entitled "Neutrophils direct preexisting matrix in to initiate repair of damaged organs" for publication in an upcoming issue of Nature Immunology.

Over the next few weeks, your paper will be copyedited to ensure that it conforms to Nature Immunology style. Once your paper is typeset, you will receive an email with a link to choose the appropriate publishing options for your paper and our Author Services team will be in touch regarding any additional information that may be required.

Please note that *Nature Immunology* is a Transformative Journal (TJ). Authors may publish their research with us through the traditional subscription access route or make their paper immediately open access through payment of an article-processing charge (APC). Authors will not be required to make a final decision about access to their article until it has been accepted. [Find out more about Transformative Journals](https://www.springernature.com/gp/open-research/transformative-journals).

Authors may need to take specific actions to achieve [compliance](https://www.springernature.com/gp/open-research/funding/policy-compliance-faqs) with funder and institutional open access mandates. For submissions from January 2021, if your research is supported by a funder that requires immediate open access (e.g. according to [Plan S principles](https://www.springernature.com/gp/open-research/plan-s-compliance)) then you should select the gold OA route, and we will direct you to the compliant route where possible. For authors selecting the subscription publication route our standard licensing terms will need to be accepted, including our [self-archiving policies](https://www.springernature.com/gp/open-research/policies/journal-policies). Those standard licensing terms will supersede any other terms that the author or any third party may assert apply to any version of the manuscript.

If you have any questions about our publishing options, costs, Open Access requirements,

or our legal forms, please contact ASJournals@springernature.com

Your paper will be published online soon after we receive your corrections and will appear in print in the next available issue. Content is published online weekly on Mondays and Thursdays, and the embargo is set at 16:00 London time (GMT)/11:00 am US Eastern time (EST) on the day of publication. Now is the time to inform your Public Relations or Press Office about your paper, as they might be interested in promoting its publication. This will allow them time to prepare an accurate and satisfactory press release. Include your manuscript tracking number (NI-A32301C) and the name of the journal, which they will need when they contact our office.

About one week before your paper is published online, we shall be distributing a press release to news organizations worldwide, which may very well include details of your work. We are happy for your institution or funding agency to prepare its own press release, but it must mention the embargo date and Nature Immunology. Our Press Office will contact you closer to the time of publication, but if you or your Press Office have any enquiries in the meantime, please contact press@nature.com.

Also, if you have any spectacular or outstanding figures or graphics associated with your manuscript - though not necessarily included with your submission - we'd be delighted to consider them as candidates for our cover. Simply send an electronic version (accompanied by a hard copy) to us with a possible cover caption enclosed.

Please note that we encourage the authors to self-archive their manuscript (the accepted version before copy editing) in their institutional repository, and in their funders' archives, six months after publication. Nature Research recognizes the efforts of funding bodies to

increase access of the research they fund, and strongly encourages authors to participate in such efforts. For information about our editorial policy, including license agreement and author copyright, please visit www.nature.com/ni/about/ed_policies/index.html

Sincerely,

Ioana Visan, Ph.D.
Senior Editor
Nature Immunology

Tel: 212-726-9207
Fax: 212-696-9752
www.nature.com/ni